# High-Order Pooling for Graph Neural Networks with Tensor Decomposition

**Chenqing Hua**[1,4*]     **Guillaume Rabusseau**[2,4,5,6]     **Jian Tang**[3,4,6]
[1]McGill University; [2]Université de Montréal; [3]HEC Montréal;
[4]Mila; [5]DIRO; [6]CIFAR AI Chair

## Abstract

Graph Neural Networks (GNNs) are attracting growing attention due to their effectiveness and flexibility in modeling a variety of graph-structured data. Exiting GNN architectures usually adopt simple pooling operations (*e.g.,* sum, average, max) when aggregating messages from a local neighborhood for updating node representation or pooling node representations from the entire graph to compute the graph representation. Though simple and effective, these linear operations do not model high-order non-linear interactions among nodes. We propose the Tensorized Graph Neural Network (tGNN), a highly expressive GNN architecture relying on tensor decomposition to model high-order non-linear node interactions. tGNN leverages the symmetric CP decomposition to efficiently parameterize permutation-invariant multilinear maps for modeling node interactions. Theoretical and empirical analysis on both node and graph classification tasks show the superiority of tGNN over competitive baselines. In particular, tGNN achieves the most solid results on two OGB node classification datasets and one OGB graph classification dataset.

## 1 Introduction

Graph neural networks (GNNs) generalize traditional neural network architectures for data in the Euclidean domain to data in non-Euclidean domains [26, 45, 34, 35]. As graphs are very general and flexible data structures and are ubiquitous in the real world, GNNs are now widely used in a variety of domains and applications such as social network analysis [20], recommender systems [49], graph reasoning [55], and drug discovery [42].

Indeed, many GNN architectures (*e.g.,* GCN [26], GAT [45], MPNN [16]) have been proposed. The essential idea of all these architectures is to iteratively update node representations by aggregating the information from their neighbors through multiple rounds of neural message passing. The final node representations can be used for downstream tasks such as node classification or link prediction. For graph classification, an additional readout layer is used to combine all the node representations to calculate the entire graph representation. In general, an effective aggregation (or pooling) function is required to aggregate the information at the level of both local neighborhoods and the entire graph. In practice, some simple aggregation functions are usually used such as sum, mean, and max. Though simple and effective in some applications, the expressiveness of these functions is limited as they only model linear combinations of node features, which can limit their effectiveness in some cases.

A recent work, principled neighborhood aggregation (PNA) [13], aims to design a more flexible aggregation function by combining multiple simple aggregation functions, each of which is associated with a learnable weight. However, the practical capacity of PNA is still limited by simply combining multiple *simple* aggregation functions. A more expressive solution would be to model high-order non-linear interactions when aggregating node features. However, explicitly modeling high-order

---

*Correspondence to: Chenqing Hua <`chenqing.hua@mail.mcgill.ca`>

non-linear interactions among nodes is very expensive, with both the time and memory complexity being exponential in the size of the neighborhood. This raises the question of whether there exists an aggregation function which can model high-order non-linear interactions among nodes while remaining computationally efficient.

In this paper, we propose such an approach based on symmetric tensor decomposition. We design an aggregation function over a set of node representations for graph neural networks, which is permutation-invariant and is capable of modeling non-linear high-order multiplicative interactions among nodes. We leverage the symmetric CANDECOMP/PARAFAC decomposition (CP) [22, 29] to design an efficient parameterization of permutation-invariant multilinear maps over a set of node representations. Theoretically, we show that the CP layer can compute any permutation-invariant multilinear polynomial, including the classical sum and mean aggregation functions. We also show that the CP layer is universally strictly more expressive than sum and mean pooling: with probability one, any function computed by a random CP layer cannot be computed using sum and mean pooling.

We propose the CP-layer as an expressive mean of performing the aggregation and update functions in GNN. We call the resulting model a *tensorized GNN* (tGNN). We evaluate tGNN on both node and graph classification tasks. Experimental results on real-world large-scale datasets show that our proposed architecture outperforms or can compete with existing state-of-the-art approaches and traditional pooling techniques. Notably, our proposed method is more effective and expressive than existing GNN architectures and pooling methods on two citation networks, two *OGB* node datasets, and one *OGB* graph dataset.

**Summary of Contributions**    We propose a new aggregation layer, the CP layer, for pooling and readout functions in GNNs. This new layer leverages the symmetric CP decomposition to efficiently parameterize polynomial maps, thus taking into account high-order multiplicative interactions between node features. We theoretically show that the CP layer can compute any permutation-invariant multilinear polynomial including sum and mean pooling. Using the CP layer as a drop-in replacement for sum pooling in classical GNN architectures, our approach achieves more effective and expressive results than existing GNN architectures and pooling methods on several benchmark graph datasets.

## 2    Preliminaries

### 2.1    Notation

We use bold font letters for vectors (*e.g.,* $\mathbf{v}$), capital letters (*e.g.,* $\mathbf{M}, \mathcal{T}$) for matrices and tensors respectively, and regular letters for nodes (*e.g.,* $v$). Let $G = (V, E)$ be a graph, where $V$ is the node set and $E$ is the edge set with self-loop. We use $N(v)$ to denote the neighborhood set of node $v$, *i.e.,* $N(v) = \{u : e_{vu} \in E\}$. A node feature is a vector $\mathbf{x} \in \mathbb{R}^F$ defined on $V$, where $\mathbf{x}_v$ is defined on the node $v$. We use $\otimes$ to denote the Kronecker product, $\circ$ to denote the outer product, and $\odot$ to denote the Hadamard (*i.e.,* component-wise) product between vectors, matrices, and tensors. For any integer $k$, we use the notation $[k] = \{1, \cdots, k\}$.

### 2.2    Tensors

We introduce basic notions of tensor algebra, more details can be found in [29]. A $k$-th order tensor $\mathcal{T} \in \mathbb{R}^{N_1 \times N_2 \times \ldots \times N_k}$ can simply be seen as a multidimensional array. The mode-$i$ fibers of $\mathcal{T}$ are the vectors obtained by fixing all indices except the $i$-th one: $\mathcal{T}_{n_1, n_2, \ldots, n_{i-1}, :, n_{i+1}, \ldots, n_k} \in \mathbb{R}^{N_i}$. The $i$-th mode matricization of a tensor is the matrix having its mode-$i$ fibers as columns and is denoted by $\mathcal{T}_{(i)}$, e.g., $\mathcal{T}_{(1)} \in \mathbb{R}^{N_1 \times N_2 \cdots N_k}$. We use $\mathcal{T} \times_i \mathbf{v} \in \mathbb{R}^{N_1 \times \cdots \times N_{i-1} \times N_{i+1} \times \cdots \times N_k}$ to denote the mode-$i$ product between a tensor $\mathcal{T} \in \mathbb{R}^{N_1 \times \cdots \times N_k}$ and a vector $\mathbf{v} \in \mathbb{R}^{N_i}$, which is defined by $(\mathcal{T} \times_i \mathbf{v})_{n_1, \ldots, n_{i-1}, n_{i+1}, \ldots, n_k} = \sum_{n_i=1}^{N_i} \mathcal{T}_{n_1, \ldots, n_k} \mathbf{v}_{n_i}$. The following useful identity relates the mode-$i$ product with the Kronecker product:

$$\mathcal{T} \times_1 \mathbf{v}_1 \times_2 \cdots \times_{k-1} \mathbf{v}_{k-1} = \mathcal{T}_{(k)} (\mathbf{v}_{k-1} \otimes \cdots \otimes \mathbf{v}_1). \tag{1}$$

### 2.3    CANDECOMP/PARAFAC Decomposition

We refer to CANDECOMP/PARAFAC decomposition of a tensor as CP decomposition [24, 22]. A Rank $R$ CP decomposition factorizes a $k$−th order tensor $\mathcal{T} \in \mathbb{R}^{N_1 \times \cdots \times N_k}$ into the sum of $R$

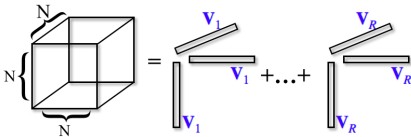

Figure 1: Example of a rank $R$ symmetric CP decomposition of a symmetric 3-order tensor $\boldsymbol{\mathcal{T}} \in \mathbb{R}^{N \times N \times N}$ such that $\boldsymbol{\mathcal{T}} = \Sigma_{r=1}^{R} \mathbf{v}_r \circ \mathbf{v}_r \circ \mathbf{v}_r$.

rank one tensors as $\boldsymbol{\mathcal{T}} = \sum_{r=1}^{R} \mathbf{v}_{1r} \circ \mathbf{v}_{2r} \circ \cdots \circ \mathbf{v}_{kr}$, where $\circ$ denotes the vector outer-product and $\mathbf{v}_{1r} \in \mathbb{R}^{N_1}, \mathbf{v}_{2r} \in \mathbb{R}^{N_2}, ..., \mathbf{v}_{kr} \in \mathbb{R}^{N_k}$ for every $r = 1, 2, ..., R$.

The decomposition vectors, $\mathbf{v}_{:r}$ for $r = 1, ..., R$, are equal in length, thus can be naturally gathered into factor matrices $\mathbf{M}_1 = [\mathbf{v}_{11}, ..., \mathbf{v}_{1R}] \in \mathbb{R}^{N_1 \times R}, ..., \mathbf{M}_k = [\mathbf{v}_{k1}, ..., \mathbf{v}_{kR}] \in \mathbb{R}^{N_k \times R}$. Using the factor matrices, we denote the CP decomposition of $\boldsymbol{\mathcal{T}}$ as

$$\boldsymbol{\mathcal{T}} = \sum_{r=1}^{R} \mathbf{v}_{1r} \circ \mathbf{v}_{2r} \circ \cdots \circ \mathbf{v}_{kr} = [\![\mathbf{M}_1, \mathbf{M}_2, ..., \mathbf{M}_k]\!].$$

The $k$-th order tensor $\boldsymbol{\mathcal{T}}$ is *cubical* if all its modes have the same size, *i.e.,* $N_1 = N_2 = ... = N_k := N$. A tensor $\boldsymbol{\mathcal{T}}$ is symmetric if it is cubical and is invariant under permutation of its indices:

$$\boldsymbol{\mathcal{T}}_{n_{\phi(1)}, ..., n_{\phi(k)}} = \boldsymbol{\mathcal{T}}_{n_1, ..., n_k}, \ n_1, \cdots, n_k \in [N]$$

for any permutation $\phi : [k] \to [k]$. A rank $R$ symmetric CP decomposition of a symmetric tensor $\boldsymbol{\mathcal{T}}$ is a decomposition of the form $\boldsymbol{\mathcal{T}} = [\![\mathbf{M}, \cdots, \mathbf{M}]\!]$ with $\mathbf{M} \in \mathbb{R}^{N \times R}$. It is well known that any symmetric tensor admits a symmetric CP decomposition [12], we illustrate a rank $R$ symmetric CP decomposition in Fig. 1.

We say that a tensor $\boldsymbol{\mathcal{T}}$ is partially symmetric if it is symmetric in a subset of its modes [28]. For example, a 3-rd order tensor $\boldsymbol{\mathcal{T}} \in \mathbb{R}^{N_1 \times N_1 \times N_3}$ is partially symmetric w.r.t. modes 1 and 2 if it has symmetric frontal slices; *i.e.,* $\boldsymbol{\mathcal{T}}_{:,:,k}$ is a symmetric matrix for all $k \in [N_3]$. We prove the fact that any partially symmetric tensor admits a partially symmetric CP decomposition in Lemma 1 in Appendix A, *e.g.,* if $\boldsymbol{\mathcal{T}} \in \mathbb{R}^{N_1 \times N_1 \times N_3}$ is partially symmetric w.r.t. modes 1 and 2, there exist $\mathbf{M} \in \mathbb{R}^{N_1 \times R}$ and $\mathbf{W} \in \mathbb{R}^{N_3 \times R}$ such that $\boldsymbol{\mathcal{T}} = [\![\mathbf{M}, \mathbf{M}, \mathbf{W}]\!]$.

## 2.4 Graph Neural Networks and Pooling Functions

Given a graph $G = (V, E)$, a graph neural network always aggregates information in a neighborhood to give node-level representations. During each message-passing iteration, the embedding $\mathbf{h}_v$ corresponding to node $v \in V$ is generated by aggregating features from $N(v)$ [18]. Formally, at the $l$-th layer of a graph neural network,

$$\mathbf{m}_{N(v)}^{(l)} = \text{AGGREGATE}^{(l)}(\{\mathbf{h}_u^{(l-1)}, \forall u \in N(v)\}), \mathbf{h}_v^{(l)} = \text{UPDATE}^{(l)}(\mathbf{h}_v^{(l-1)}, \mathbf{m}_{N(v)}^{(l)}), \quad (2)$$

where $\text{AGGREGATE}^{(l)}(\cdot)$ and $\text{UPDATE}^{(l)}(\cdot)$ are differentiable functions, the former being permutation-invariant. In words, $\text{AGGREGATE}^{(l)}(\cdot)$ first aggregates information from $N(v)$, then $\text{UPDATE}^{(l)}(\cdot)$ combines the aggregated message and previous node embedding $\mathbf{h}_v^{(l-1)}$ to give a new embedding.

The node representation $\mathbf{h}_v^{(L)}$ of node $v$ from the last GNN layer $L$ can be used for predicting relevant properties of the node. For graph classification, an additional $\text{READOUT}(\cdot)$ function aggregates node representations from the final layer to obtain a graph representation $\mathbf{h}_G$ of graph $G$ as,

$$\mathbf{h}_G = \text{READOUT}(\{\mathbf{h}_v^{(L)} | v \in V\}), \quad (3)$$

where $\text{READOUT}(\cdot)$ can be a simple permutation-invariant function (*e.g.,* sum, mean, *etc.*) or a more sophisticated graph pooling function [50, 48].

In the general design of GNNs, it is important to have an effective pooling function to aggregate messages from local neighborhoods and update node representations (see Eq. equation 2), and to combine node representations to compute a representation at the graph level (see Eq. equation 3).

# 3 Meaning of High-Order of tGNN

In our paper, high-order refers to multi-dimensional feature products between nodes in a neighborhood. [38, 43] are previous works on high-dimensional feature products, and introduce how to use tensor methods to achieve those products. They have stated the importance of having high-order products in physics and geometries. For example, if we have $x_1$ and $x_2$, a simple GNN layer can result in a vector of $[x_1, x_2, x_1 + x_2]$, but our high-order layer result in a vector of $[x_1, x_2, x_1 + x_2, x_1 x_2]$ with another term $x_1 x_2$. And if we have we have $x_1$, $x_2$, and $x_3$, a simple low-order GNN layer will result in $[x_1, x_2, x_3, x_1 + x_2, x_1 + x_3, x_2 + x_3, x_1 + x_2 + x_3]$ while the high-order layer will result in a vector of $[x_1, x_2, x_3, x_1 + x_2, x_1 + x_3, x_2 + x_3, x_1 + x_2 + x_3, x_1 x_2, x_1 x_3, x_2 x_3, x_1 x_2 x_3]$ with extra 4 terms at a higher feature dimension. In general content of GNNs, high-order will normally refer to the ability of capturing long-range dependencies, the interactions between long-range nodes [48, 2, 46], but ours refers to the multi-dimensional node feature products. However, the CP layer is able to capture stacking long-range dependencies by stacking layers like other simple GNNs [26, 45].

One may raise a question on why we need to compute multi-dimensional feature products. The idea is fairly easy if one thinks the multi-dimensional feature space. Take a molecular graph dataset as an example, and blood pressure could be one feature. Many studies analyze the impact of blood pressure on one's health. Blood pressure is the feature in this case. So, one could try to predict life expectancy using blood pressure as a feature, but you can imagine that this might not be sufficient. You really need more information. So you could add age, exercise frequency, and body fat to see if you can predict a person's life expectancy. Those additional measurements are also features and now every individual has a multi-dimensional feature vector consisting of these measurements. However, a simple GNN with one-dimensional feature interactions may not capture the correlation between age, exercise frequency, and body fat. There could be some correlation between the three factors to predict life expectancy. A simple GNN that does age+exercise frequency+body fat is not able to capture the correlation between those factors. Still, our multi-dimensional feature product can capture age×exercise frequency, age×body fat, exercise frequency×body fat, age×exercise frequency×body fat, all the correlations to predict life expectancy. So in molecular graph case, high-dimensional feature products are importantly needed to make a prediction as most features are correlated (like in atom graphs, atom charge, and atom mass), and our model shows some significant improvements on molecular datasets in Sec.6. And the high-dimensional feature products might also be important for other networks and applications [38, 43].

# 4 Tensorized Graph Neural Network

In this section, we introduce the CP-layer and tensorized GNNs (tGNN). For convenience, we let $\{\mathbf{x}_1, \mathbf{x}_2, ..., \mathbf{x}_k\}$ denote features of a node $v$ and its 1-hop neighbors $N(v)$ such that $|\{v\} \cup N(v)| = k$.

## 4.1 Motivation and Method

We leverage the symmetric CP decomposition to design an efficient parameterization of permutation-invariant multilinear maps for aggregation operations in graph neural networks, Tensorized Graph Neural Network (tGNN), resulting in a more expressive high-order node interaction scheme. We visualize the CP pooling layer and compare it with sum pooling in Fig. 2.

Let $\mathcal{T} \in \mathbb{R}^{N \times N \times \cdots \times N \times M}$ of order $k + 1$ be a tensor which is partially symmetric w.r.t. its first $k$ modes. We can parameterize $\mathcal{T}$ using a rank $R$ partially symmetric CP decomposition (see Section 2.3): $\mathcal{T} = [\![\mathbf{W}, \cdots, \mathbf{W}, \mathbf{M}]\!]$ where $\mathbf{W} \in \mathbb{R}^{N \times R}$ and $\mathbf{M} \in \mathbb{R}^{M \times R}$. Such a tensor naturally defines a map from $(\mathbb{R}^N)^k$ to $\mathbb{R}^M$ using contractions over the first $k$ modes:

$$f(\mathbf{x}_1, \cdots, \mathbf{x}_k) = \mathcal{T} \times_1 \mathbf{x}_1 \times_2 \cdots \times_k \mathbf{x}_k = [\![\underbrace{\mathbf{W}, \cdots, \mathbf{W}}_{k \text{ times}}, \mathbf{M}]\!] \times_1 \mathbf{x}_1 \times_2 \cdots \times_k \mathbf{x}_k. \quad (4)$$

This map satisfies two very important properties for GNNs: it is *permutation-invariant* (due to the partial symmetry of $\mathcal{T}$) and *its number of parameters is independent of $k$* (due to the partially symmetric CP parameterization). Thus, using only two parameter matrices of fixed size, the map in Eq. equation 4 can be applied to sets of $N$-dimensional vectors of arbitrary cardinality. In particular, we will show that it can be leveraged to replace both the AGGREGATE and UPDATE functions in GNNs.

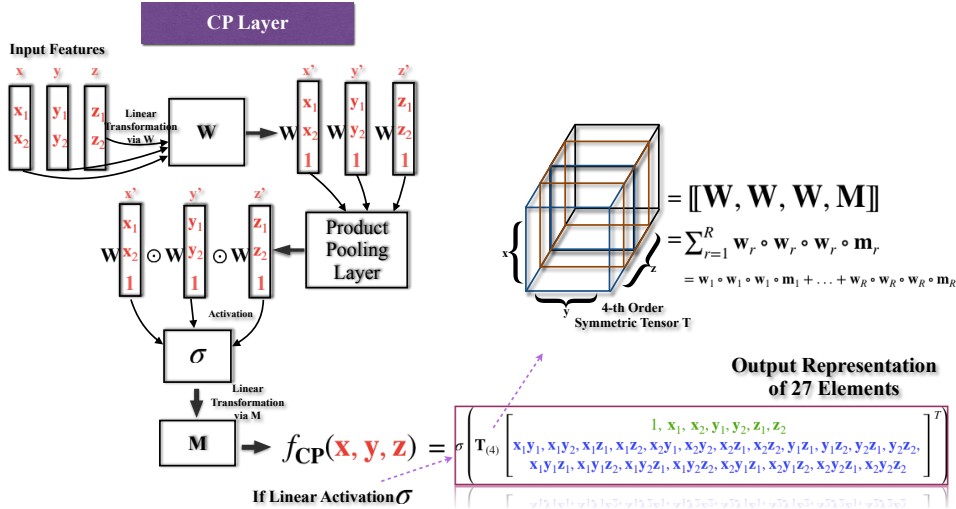

Figure 2: (Left) Sum pooling followed by a FC layer: the output takes individual components of the input into account. (Right) The CP layer can be interpreted as a combination of product pooling with linear layers (with weight matrices $\mathbf{W}$ and $\mathbf{M}$) and non-linearities. The weight matrices of a CP layer corresponds to a partially symmetric CP decomposition of a weight tensor $\mathcal{T} = [\![\mathbf{W}, \mathbf{W}, \mathbf{W}, \mathbf{M}]\!]$. It shows that the output of a CP layer takes high-order multiplicative interactions of the inputs' components into account (in contrast with sum pooling that only considers 1st order terms).

There are several way to interpret the map in Eq. equation 4. First, from Eq. equation 1 we have

$$f(\mathbf{x}_1, \cdots, \mathbf{x}_k) = \mathcal{T} \times_1 \mathbf{x}_1 \times_2 \cdots \times_k \mathbf{x}_k = \mathcal{T}_{(k+1)}(\mathbf{x}_k \otimes \mathbf{x}_{k-1} \otimes \cdots \otimes \mathbf{x}_1),$$

where $\mathcal{T}_{(k+1)} \in \mathbb{R}^{M \times N^k}$ is the mode-$(k+1)$ matricization of $\mathcal{T}$. This shows that each element of the output $f(\mathbf{x}_1, \cdots, \mathbf{x}_k)$ is a linear combinations of terms of the form $(\mathbf{x}_1)_{i_1}(\mathbf{x}_2)_{i_2} \cdots (\mathbf{x}_k)_{i_k}$ ($k$-th order multiplicative interactions between the components of the vectors $\mathbf{x}_1, \cdots, \mathbf{x}_k$). That is, $f$ is a multivariate polynomial map of order $k$ involving only $k$-th order interactions. By using homogeneous coordinates, *i.e.,* appending an entry equal to one to each of the input tensors $\mathbf{x}_i$, the map $f$ becomes a more general polynomial map taking into account all multiplicative interactions between the $\mathbf{x}_i$ *up to* the $k$-th order:

$$f(\mathbf{x}_1, \cdots, \mathbf{x}_k) = \mathcal{T} \times_1 \begin{bmatrix} \mathbf{x}_k \\ 1 \end{bmatrix} \times_2 \cdots \times_k \begin{bmatrix} \mathbf{x}_1 \\ 1 \end{bmatrix} = \mathcal{T}_{(k+1)}\left( \begin{bmatrix} \mathbf{x}_k \\ 1 \end{bmatrix} \otimes \cdots \otimes \begin{bmatrix} \mathbf{x}_1 \\ 1 \end{bmatrix} \right)$$

where $\mathcal{T}$ is now of size $(N+1) \times \cdots \times (N+1) \times M$ and can still be parameterized using the partially symmetric CP decomposition $\mathcal{T} = [\![\mathbf{W}, \cdots, \mathbf{W}, \mathbf{M}]\!]$ with $\mathbf{W} \in \mathbb{R}^{(N+1) \times R}$ and $\mathbf{M} \in \mathbb{R}^{M \times R}$. With this parameterization, one can check that

$$f(\mathbf{x}_1, \cdots, \mathbf{x}_k) = \mathbf{M}\left( \left( \mathbf{W}^\top \begin{bmatrix} \mathbf{x}_1 \\ 1 \end{bmatrix} \right) \odot \cdots \odot \left( \mathbf{W}^\top \begin{bmatrix} \mathbf{x}_k \\ 1 \end{bmatrix} \right) \right)$$

where $\odot$ denotes the component-wise product between vectors. The map $f$ can thus be seen as the composition of a linear layer with weight $\mathbf{W}$, a multiplicative pooling layer, and another linear map $\mathbf{M}$. Since it is permutation-invariant and can be applied to any number of input vectors, this map can be used as both the aggregation, update, and readout functions of a GNN using non-linear activation functions, which leads us to introduce the novel *CP layer* for GNN.

**Definition 1.** *(CP layer) Given parameter matrices $\mathbf{M} \in \mathbb{R}^{d \times R}$ and $\mathbf{W} \in \mathbb{R}^{F+1 \times R}$ and activation functions $\sigma, \sigma'$, a rank $R$ CP layer computes the function $f_{CP} : \cup_{i \geq 1}(\mathbb{R}^F)^i \to \mathbb{R}^d$ defined by*

$$f_{\mathbf{CP}}(\mathbf{x}_1, \cdots, \mathbf{x}_k) = \sigma'\left( \mathbf{M}\left( \sigma\left( \mathbf{W}^\top \begin{bmatrix} \mathbf{x}_1 \\ 1 \end{bmatrix} \odot \cdots \odot \mathbf{W}^\top \begin{bmatrix} \mathbf{x}_k \\ 1 \end{bmatrix} \right) \right) \right)$$

*for any $k \geq 1$ and any $\mathbf{x}_1, \cdots, \mathbf{x}_k \in \mathbb{R}^F$.*

The rank $R$ of a CP layer is a hyperparameter controlling the trade-off between parameter efficiency and expressiveness. Note that the CP layer computes AGGREGATE and UPDATE (see Eq. 2) in one step. One can think of the component-wise product of the $\mathbf{W}^\top[\mathbf{x}_i\ 1]^\top$ as AGGREGATE, while

the UPDATE corresponds to the two non-linear activation functions and linear transformation $\mathbf{M}$. We observed in our experiments that the non-linearity $\sigma$ is crucial to avoid numerical instabilities during training caused by repeated products of $\mathbf{W}$. In practice, we use *Tanh* for $\sigma$ and *ReLU* for $\sigma'$. Fig. 2 graphically explains the computational process of a CP layer, comparing it with a classical sum pooling operation. We intuitively see in this figure that the CP layer is able to capture high order multiplicative interactions that are not modeled by simple aggregation functions such as the sum or the mean. In the next section, we theoretically formalize this intuition.

**Complexity Analysis** The sum, mean and max poolings result in $O(F_{in}(N + F_{out}))$ time complexity, while CP pooling is $O(R(NF_{in} + F_{out}))$, where $N$ denotes the number of nodes, $F_{in}$ is the input feature dimension, $F_{out}$ is out feature dimension, and $R$ is the CP decomposition rank. In Sec. 6.3, we experimentally compare tGNN and CP pooling with various GNNs and pooling techniques to show the model efficiency with limited computation and time budgets.

## 4.2 Theoretical Analysis

We now analyze the expressive power of CP layers. In order to characterize the set of functions that can be computed by CP layers, we first introduce the notion of multilinear polynomial. A multilinear polynomial is a special kind of vector-valued multivariate polynomial in which no variables appears with a power of 2 or higher. More formally, we have the following definition.

**Definition 2.** *A function $g : \mathbb{R}^k \to \mathbb{R}$ is called a* univariate multilinear polynomial *if it can be written as*

$$g(a_1, a_2, \cdots, a_k) = \sum_{i_1=0}^{1} \cdots \sum_{i_k=0}^{1} \tau_{i_1 i_2 \cdots i_k} a_1^{i_1} a_2^{i_2} \cdots a_k^{i_k}$$

*where each $\tau_{i_1 i_2 \cdots i_k} \in \mathbb{R}$. The* degree *of a univariate multilinear polynomial is the maximum number of distinct variables occurring in any of the non-zero monomials $\tau_{i_1 i_2 \cdots i_n} a_1^{i_1} a_2^{i_2} \cdots a_k^{i_k}$.*

*A function $f : (\mathbb{R}^d)^k \to \mathbb{R}^p$ is called a* multilinear polynomial map *if there exist univariate multilinear polynomials $g_{i,j_1,\cdots,j_n}$ for $j_1, \cdots, j_k \in [d]$ and $i \in [p]$ such that*

$$f(\mathbf{x}_1, \cdots, \mathbf{x}_k)_i = \sum_{j_1,\cdots,j_k=1}^{d} g_{i,j_1,\cdots,j_k}((\mathbf{x}_1)_{j_1}, \cdots, (\mathbf{x}_k)_{j_k})$$

*for all $\mathbf{x}_1, \cdots, \mathbf{x}_k \in \mathbb{R}^d$ and all $i \in [p]$. The degree of $f$ is the highest degree of the multilinear polynomials $g_{i,j_1,\cdots,j_k}$.*

The following theorem shows that CP layers can compute any permutation-invariant multilinear polynomial map. We also visually represent the expressive power of CP layers in Fig. 3, showing that the class of functions computed by CP layer subsumes multilinear polynomials (including sum and mean aggregation functions).

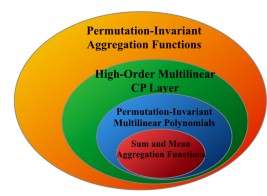

**Theorem 1.** *The function computed by a CP layer (Eq. equation 1) is permutation-invariant. In addition, any* permutation-invariant *multilinear polynomial $f : (\mathbb{R}^F)^k \to \mathbb{R}^d$ can be computed by a CP layer (with a linear activation function).*

Note also that in Fig. 3 the CP layer is strictly more expressive than permutation invariant multilinear polynomials due to the non-linear activation functions in Def. 1. Since the classical sum and mean pooling aggregation functions are degree 1 multilinear polynomial maps, it readily follows from the previous theorem that the CP layer is more expressive than these standards aggregation functions. However, it is natural to ask how many parameters a CP layer needs to compute sums and means. We answer this question in the following theorem.

Figure 3: Visualization of relations of {permutation-invariant function space} $\supseteq$ {CP function space} $\supseteq$ {permutation-invariant multilinear polynomial space} $\supseteq$ {sum and mean aggregation functions}.

**Theorem 2.** *A CP layer of rank $F \cdot k$ can compute the sum and mean aggregation functions over $k$ vectors in $\mathbb{R}^F$.*

*Consequently, for any $k \geq 1$ and any GNN $\mathcal{N}$ using mean or sum pooling with feature and embedding dimensions bounded by $F$, there exists a GNN with CP layers of rank $F \cdot k$ computing the same function as $\mathcal{N}$ over all graphs of uniform degree $k$.*

It follows from this theorem that a CP layer with $2F^2k$ can compute sum and mean aggregation over sets of $k$ vectors. While Theorem 2 shows that any function using sum and mean aggregation can be computed by a CP layer, the next theorem shows that the converse is not true, *i.e.,* the CP layer is a strictly more expressive aggregator than using the mean or sum.

**Theorem 3.** *With probability one, any function $f_{CP} : (\mathbb{R}^F)^k \to \mathbb{R}^d$ computed by a CP layer (of any rank) whose parameters are drawn randomly (from a distribution which is continuous w.r.t. the Lebesgue measure) cannot be computed by a function of the form*

$$g_{sum} : \mathbf{x}_1, \cdots, \mathbf{x}_k \mapsto \sigma' \left( \mathbf{M} \left( \sigma \left( \sum_{i=1}^{k} \mathbf{W}^\top \mathbf{x}_i \right) \right) \right)$$

*where $\mathbf{M} \in \mathbb{R}^{d \times R}$, $\mathbf{W} \in \mathbb{R}^{F \times R}$ and $\sigma$, $\sigma'$ are component-wise activation function.*

This theorem not only shows that there exist functions computed by CP layers that cannot be computed using sum pooling, but that this is the case for *almost all* functions that can be computed by (even rank-one) CP layers.

From an expressive power viewpoint, we showed that a CP layer is able to leverage both low and high-order multiplicative interactions. However, from a learning perspective, it is clear that the CP layer has a natural inductive bias towards capturing high-order interactions. We are not enforcing any sparsity in the tensor parameterizing the polynomial, thus the number and magnitude of weights corresponding to high-order terms will dominate the result (intuitively, learning a low order polynomial would imply setting most of these weights to zero). In order to counterbalance this bias, we complement the CP layer with simple but efficient linear low-order interactions (reminiscent of the idea behind residual networks [21]) when using it in tGNN:

$$f(\mathbf{x}_1, \cdots, \mathbf{x}_k) = \sigma' \left( \mathbf{M} \left( \sigma \left( \mathbf{W}_1^\top \begin{bmatrix} \mathbf{x}_1 \\ 1 \end{bmatrix} \odot \cdots \odot \mathbf{W}_1^\top \begin{bmatrix} \mathbf{x}_k \\ 1 \end{bmatrix} \right) \right) \right) + \sigma''(\mathbf{W}_2^\top \mathbf{x}_1 + ... + \mathbf{W}_2^\top \mathbf{x}_k) \quad (5)$$

where the first term corresponds to the CP layer and the second one to a standard sum pooling layer (with $\sigma$, $\sigma'$ and $\sigma''$ being activation functions).

## 5 Related Work and Discussion

We now discuss relevant work on the parameterization of tensors on graph neural networks. In general, our work relates to three areas of deep learning: (1) aggregation functions, (2) universal approximator for set aggregation, (3) high-order pooling, and (4) tensor methods for deep learning.

**GNN & Aggregation Scheme** [26] successfully define convolutions on graph-structured data by averaging node information in a neighborhood. [48] prove the incomplete expressivity of mean aggregation to distinguish nodes, and further propose to use sum aggregation to differentiate nodes with similar properties. [13] further generalize this idea and show that mean aggregation can be a particular case of sum aggregation with a linear multiplier, and further propose an architecture with multiple aggregation channels to adaptively learn low-order information. Most GNNs use low-order aggregation schemes for learning node representations. To the best of our knowledge, tGNN is the first GNN architecture that adopts high-order multiplicative interactions in aggregation.

**Universal Approximator & Permutation-invariant NN** Universal approximators for set aggregation functions have been previously proposed and studied. [51] show that sum pooling is enough provided that it is combined with two universal approximators. [48] further discuss the limitations of non-injective set function. [25] propose a generalization of transformers to permutation-invariant sets. Most universal approximation results for GNN relies on combining simple aggregation functions (e.g. sum, mean) with universal approximators for the feature and output maps. In contrast, the CP layer achieves the same goal of being an universal approximator but using a different mean: explicit computation of multilinear polynomials (in an effective manner using the CP parameterization). While the approaches might be expressively similar, they may differ from a learning aspect, *i.e.,* how easy it is for these models to learn a function from data.

**High-order Pooling** [2] formulate the pooling problem as a multiset encoding problem with auxiliary information about the graph structure, and propose an attention-based pooling layer that captures the interaction between nodes according to their structural dependencies. [46] apply second-order statistic methods because the use of second-order statistics takes advantage of the Riemannian

geometry of the space of symmetric positive definite matrices. However, high-order in CP pooling refers to high-dimensional multiplicative feature products, which is different than previous literature.

**Tensor Methods**   Tensor methods allow one to define meaningful geometries to build more expressive models. Authors in [52, 30] define geometries of tensor networks on complex or hypercomplex manifolds, their models encompass greater freedom in the choice of the product between the algebra elements. [7] develop a general framework for both probabilistic and neural models for tree-structured data with a tensor-based aggregation function. [6] leverages permutation-invariant CP-based aggregation function to capture high-order interactions in NLP tasks. Using multiplicative interactions as a powerful source of non-linearities in neural network models have been studied previously for convolutional [11, 10, 32] and recurrent networks [41, 44, 47]. The connection between such multiplicative interactions with tensor networks have been leveraged both from theoretical and practical perspectives. The CP layer can be seen as a graph generalization of the convolutional and recurrent arithmetic circuits considered in [10] and [31], respectively.

## 6   Experiments on Real-World Datasets

In this section, we evaluate Tensorized Graph Neural Net on real-world node- and graph-level datasets. We introduce experiment setup in 6.1, compare tGNN with the state-of-the-arts models in 6.2, and conduct ablation study on model performance and efficiency in 6.3. The hyperparameter and computing resources are attached in Appendix K. Dataset information can be found in Appendix J.

### 6.1   Experiment Setup

In this work, we conduct experiments on three citation networks (*Cora*, *Citeseer*, *Pubmed*) and three *OGB* datasets (*PRODUCTS*, *ARXIV*, *PROTEINS*) [23] for node-level tasks, one *OGB* dataset (*MolHIV*) [23] and three *benchmarking* datasets (*ZINC*, *CIFAR10*, *MNIST*) [14] for graph-level tasks.

**Training Procedure**   For three citation networks (*Cora*, *Citeseer*, *Pubmed*), we run experiments 10 times on each dataset with 60%/20%/20% random splits used in [9], and report results in Tab. 1. For data splits of *OGB* node and graph datasets, we follow [23], run experiments 5 times on each dataset (due to training cost), and report results in Tab. 1, 2. For *benchmarking* datasets, we run experiments 5 times on each dataset with data split used in [14], and report results in Tab. 2. To avoid numerical instability and floating point exception in tGNN training, we sample 5 neighbors for each node. For graph datasets, we do not sample because the training is already in batch thus numerical instability can be avoided, and we apply the CP pooling at both node-level aggregation and graph-level readout.

**Model Comparison**   tGNN has two hyperparameters, hidden unit and decomposition rank, we fix hidden unit and explore decomposition rank. For citation networks, we compare 2-layer GNNs with 32 hidden units. And for *OGB* and *benchmarking* datasets, we use 32 hidden units for tGNN, and the results for all other methods are reported from the leaderboards and corresponding references.

Particularly, tGNN and CP pooling are more effective and expressive than existing pooling techniques for GNNs on two citation networks, two *OGB* node datasets, and one *OGB* graph dataset in Tab. 1, 2.

### 6.2   Real-world Datasets

In this section, we present tGNN performance on node- and graph-level tasks. We compare tGNN with several classic baseline models under the same training setting. For three citation networks, we compare tGNN with several baselines including GCN [26], GAT [45], GraphSAGE [19], $H_2$GCN [54], GPRGNN [9], APPNP [27] and MixHop [1]; for three *OGB* node datasets, we compare tGNN with MLP, Node2vec [17], GCN [26], GraphSAGE [19] and DeeperGCN [33]. And for graph-level tasks, we compare tGNN with several baselines including MLP, GCN [26], GIN [48], DiffPool [50], GAT [45], MoNet [37], GatedGCN [4], PNA [13], PHMGNN [30] and DGN [3]. The model choice is because we propose a new pooling method and want to mainly compare tGNN with other poolings in standard GNN architectures. Current leading models on OGB leaderboards [23] adopt transformer, equivariant, fingerprint, C&S, or others, which are more complex than standard GNNs. We visualize the performance boost and comparisons with GNNs and pooling techniques in Tab. 1,2.

From Tab. 1, we can observe that tGNN outperforms all classic baselines on *Cora*, *Pubmed*, *PRODUCTS* and *ARXIV*, and have slight improvements on the other datasets but underperforms GCN on *Citeseer* and DeeperGCN on *PROTEINS*. On the citation networks, tGNN outperforms others on 2 out of 3 datasets. Moreover, on the *OGB* node datasets, even when tGNN is not ranked first, it is still very competitive (top 3 for all datasets). We believe it is reasonable and expected that tGNN does not outperform all methods on all datasets. But overall tGNN shows very competitive performance and

Table 1: Results of node-level tasks. **Left Table**: tGNN in comparison with GNN architectures on citation networks. **Right Table**: tGNN in comparison with GNN architectures on *OGB* datasets.

| DATASET MODEL | Cora Acc | Citeseer Acc | Pubmed Acc |
|---|---|---|---|
| GCN | 0.8778±0.0096 | 0.8139±0.0123 | 0.8890±0.0032 |
| GAT | 0.8686±0.0042 | 0.6720±0.0046 | 0.8328±0.0012 |
| GraphSAGE | 0.8658±0.0026 | 0.7624±0.0030 | 0.8658±0.0011 |
| H$_2$GCN | 0.8752±0.0061 | 0.7997±0.0069 | 0.8778±0.0028 |
| GPRGNN | 0.7951±0.0036 | 0.6763±0.0038 | 0.8507±0.0009 |
| APPNP | 0.7941±0.0038 | 0.6859±0.0030 | 0.8502±0.0009 |
| MixHop | 0.6565±0.1131 | 0.4952±0.1335 | 0.8704±0.0410 |
| tGNN | 0.8808±0.0131 | 0.80.51±0.0192 | 0.9080±0.0018 |

| DATASET MODEL | PRODUCTS Acc | ARXIV Acc | PROTEINS AUC |
|---|---|---|---|
| MLP | 0.6106±0.0008 | 0.5550±0.0023 | 0.7204±0.0048 |
| Node2vec | 0.7249±0.0010 | 0.7007±0.0013 | 0.6881±0.0065 |
| GCN | 0.7564±0.0021 | 0.7174±0.0029 | 0.7251±0.0035 |
| GraphSAGE | 0.7850±0.0016 | 0.7149±0.0027 | 0.7768±0.0020 |
| DeeperGCN | 0.8098±0.0020 | 0.7192±0.0016 | 0.8580±0.0017 |
| tGNN | 0.8179±0.0054 | 0.7538±0.0015 | 0.8255±0.0049 |

deliver significant improvement on challenging graph benchmarks compared to popular commonly used pooling methods (with comparable computational cost).

Table 2: Results of tGNN on graph-level tasks in comparison with GNN architectures.

| DATASET | | ZINC No edge features MAE | CIFAR10 No edge features Acc | MNIST No edge features Acc | MolHIV No edge features AUC |
|---|---|---|---|---|---|
| Dwivedi et al. and Hu et al. | MLP | 0.710±0.001 | 0.560±0.009 | 0.945±0.003 | |
| | GCN | 0.469±0.002 | 0.545±0.001 | 0.899±0.002 | 0.761±0.009 |
| | GIN | 0.408±0.008 | 0.533±0.037 | 0.939±0.013 | 0.756±0.014 |
| | DiffPool | 0.466±0.006 | 0.579±0.005 | 0.950±0.004 | |
| | GAT | 0.463±0.002 | 0.655±0.003 | 0.956±0.001 | |
| | MoNet | 0.407±0.007 | 0.534±0.004 | 0.904±0.005 | |
| | GatedGCN | 0.422±0.006 | 0.692±0.003 | 0.974±0.001 | |
| Corso et al. | PNA | 0.320±0.032 | 0.702±0.002 | 0.972±0.001 | 0.791±0.013 |
| Le et al. | PHM-GNN | | | | 0.793±0.012 |
| Beaini et al. | DGN | 0.219±0.010 | 0.727±0.005 | | 0.797±0.009 |
| Ours | tGNN | 0.301±0.008 | 0.684±0.006 | 0.965±0.002 | 0.799±0.016 |

Table 3: Results of the tGNN ablation study on two node- and one graph-level tasks. tGNN in comparison with high-order CP pooling and low-order linear sum pooling.

| DATASET MODEL | Cora Acc | Pubmed Acc | ZINC MAE |
|---|---|---|---|
| Non-Linear CP Pooling | 0.8655±0.0375 | 0.8679±0.0103 | 0.407±0.025 |
| Linear Sum Pooling | 0.8623±0.0107 | 0.8531±0.0009 | 0.440±0.010 |
| Non-Linear CP + Linear Sum | 0.8780±0.0158 | 0.9018±0.0015 | 0.301±0.008 |

In Tab. 2, we present tGNN performance on graph property prediction tasks. tGNN achieves state-of-the-arts results on *MolHIV*, and have slight improvements on other three *Benchmarking* graph datasets. Overall tGNN achieves more effective and accurate results on 5 out of 10 datasets comparing with existing pooling techniques, which suggests that high-order CP pooling can leverage a GNN to generalize better node embeddings and graph representations.

## 6.3 Ablation and Efficiency Study

In the ablation study, we first investigate the effectiveness of having the high-order non-linear CP pooling and adding the linear low-order transformation in Tab. 3, then investigate the relations of the model performance, efficiency, and tensor decomposition rank in Fig. 4. Moreover, we compare tGNN with different GNN architectures and aggregation functions to show the efficiency in Tab. 5, 7 by showing the number of model parameters, computation time, and accuracy.

In Tab. 3, we test each component, high-order non-linear CP pooling, low-order linear sum pooling, and two pooling techniques combined, separately. We fix 2-layer GNNs with 32 hidden unit and 64 decomposition rank, and run experiments 10 times on *Cora* and *Pubmed* with 60%/20%/20% random splits used in [9], run *ZINC* 5 times with 10,000/1,000/1,000 graph split used in [14].

From the results, we can see that adding the linear low-order interactions helps put essential weights on them. Ablation results show that high-order CP pooling has the advantage over low-order linear pooling for generating expressive node and graph representations, moreover, tGNN is more expressive with the combination of high-order pooling and low-order aggregation. This illustrates the necessity of learning high-order components and low-order interactions simultaneously in tGNN.

**Computational Aspect** In Fig. 4, we compare model performance with computation costs. We fix a 2-layer tGNN with 32 hidden channels with 8,16,...,2048,4096 rank, and 2-layer baselines with 8,16,...,2048,4096 hidden dim. We run 10 times on each citation network with the same 60%/20%/20% random splits for train/validation/test and draw the relations of average test accuracy, rank/hidden dim, and computation costs. From the figure, we can see that the model performance can be improved with higher ranks (*i.e.,* Tensor $\mathcal{T}$ is more accurately computed as the rank $R$ gets larger), but training time is also increased, thus it is a trade-off between classification accuracy and

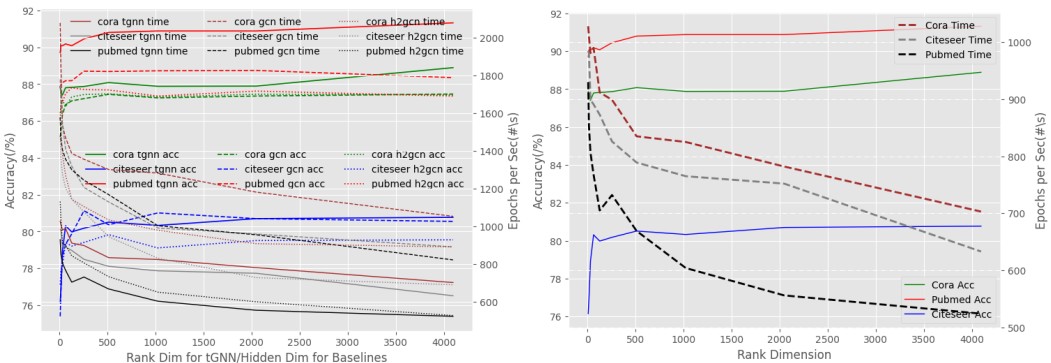

Figure 4: Results of node classification with increasing rank dimension on three citation datasets. **Left Figure**: Left axis shows accuracy, right axis shows #training epochs second, and horizontal axis indicates rank dim of tGNN or hidden dim of baselines. **Right Figure**: Left axis shows accuracy, right axis shows #training epochs second, and horizontal axis indicates rank dim of tGNN.

computation efficiency. And in comparison with baselines, tGNN still has marginal improvements with higher ranks while the baselines stop improving with larger hidden dimensions.

In Sec. 4.1, we theoretically discuss the time complexity of tGNN. In Appendix H, we experimentally assess model efficiency by comparing tGNN with GCN [26], GAT [45], GCN2 [8], and mean, max poolings on *Cora* on a CPU over 10 runtimes, and compare the number of model parameters, training epochs per second, and accuracy. The experiments show that tGNN is more competitive in terms of running time and better accuracy with a fixed number of parameters and the same time budget.

# 7 Conclusion and Future Work

In this paper, we theoretically develop a high-order permutation-invariant multilinear map for node aggregation and graph pooling via tensor parameterization. We show its powerful ability to compute any permutation-invariant multilinear polynomial including sum and mean pooling functions. Experiments demonstrate that tGNN is more effective and accurate on 5 out of 10 datasets, showcasing the relevance of tensor methods for high-order graph neural network models.

For future work, one interesting direction is to augment various GNN architectures with non-linear high-order CP layers. Most of the existing GNN models adopt low-order aggregation and pooling functions, it would be interesting to equip current state-of-the-art models with high-order pooling functions for a potential performance boost. Another interesting direction is to enhance tGNN with an adaptive channel mixing mechanism. In tGNN with high-order and low-order pooling functions, one node receives two channels of information combined in a linear way. The linear combination may not be sufficient to balance and extract high-order components and low-order components, so it would be interesting to design an adaptive channel mixing tGNN model for learning from different node-wise components.

### Acknowledgements

This research was supported by the Canadian Institute for Advanced Research (CIFAR AI chair program), the Natural Sciences and Engineering Research Council of Canada (Discovery program, RGPIN-2019-05949). This research was enabled in part by support provided by Calcul Québec (www.calculquebec.ca) and the Digital Research Alliance of Canada (alliance.can.ca). The authors acknowledge the material support of NVIDIA in the form of computational resources.

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
