# OpenReview forum: "High-Order Pooling for Graph Neural Networks with Tensor Decomposition"
_NeurIPS.cc/2022/Conference — NeurIPS 2022 Accept_

### Official Review · Reviewer_1gCd · 2022-06-23

**Rating:** 6
**Confidence:** 4
**Soundness:** 2 fair
**Presentation:** 4 excellent
**Contribution:** 2 fair

**Summary:**

This paper proposes to consider non-linear high-order multiplicative interactions among nodes with symmetric tensor decomposition, for node aggregations in graph neural networks (GNNs). In particular,
* The authors motivate that existing GNNs rely on naive pooling operations, such as sum or mean, which are insufficient for aggregating nodes for obtaining node- and graph-level representations.
* The authors then propose to consider high-order interactions among nodes with the existing symmetric CP decomposition scheme, and the authors show that the proposed one can approximate permutation-invariant multilinear polynomial including sum and mean aggregations.
* The authors verify their model on node and graph classification benchmark datasets, on which the proposed method named tGNN outperforms existing baselines on most datasets.

**Questions:**

### Major Questions and Suggestions
* Regarding analyses on the model complexity, I suggest authors increase the number of sampled nodes (currently, they are too small, for example, 3 or 5) and then compare the complexity across different models in this setting of increased sample sizes.
* Also, regarding analyses on the model complexity, I suggest the authors include the complexities of baselines in Figure 4 of the main paper, and then discuss the complexity by comparing the proposed tGNN and baselines.
* There are two related works: 1) There is a relevant work that considers interactions among nodes on graph pooling, which is also a universal approximator for set aggregation [1]. 2) There is a relevant work that proposes second-order pooling for graph neural networks [2]. Those works should be discussed and compared on the graph classification task.
* In Table 1, why the most basic GCN outperforms all the other recent GNNs, which is quite weird?
* Also, it is quite weird that the performance results in Table 1 and Table 4 are different.
* I suggest authors empirically verify that the CP layer can approximate sum and mean pooling functions. While the authors theoretically analyze that the proposed CP layer can compute them, I am not certain this is empirically possible, as the CP layer tends to capture high-order interactions as in Line 202 while ignoring low-order interactions.
* Also, similarly to the above question, the evidence of tenting to capture high-order interactions and why not exist. I would like to see how much and why the CP layer prefers high-order interactions.
* In Figure 2, which part is the illustration of sum pooling? It seems there is only the illustration of the proposed CP layer, and the sum pooling is not illustrated but mentioned in the caption.

### Minor Questions and Suggestions
* This is a simple question that is the linear sum pooling in Table 3 the model of equation (5) without using the left (first) term?
* The number of sampled nodes is not consistent across different tables: in Line 263, the authors sample 5 neighbors, however, in Table 4, the authors sample 3 neighbors. Why does there exists inconsistency?
* In Line 288, the description is not matched to the results in Table 1: there are no datasets that the proposed tGNN is ranked at four.
* I cannot see the text in Figure 3 clearly, due to its background color.

### Typos
* In line 154, such at -> such as.

---

[1] Accurate Learning of Graph Representations with Graph Multiset Pooling. ICLR 2021.

[2] Second-Order Pooling for Graph Neural Networks. IEEE Transactions on Pattern Analysis and Machine Intelligence, 2020

---

**Update after the author-reviewer interaction phase**: thanks authors for additionally revising the manuscript. Since I cannot write public comments that the authors can see anymore, I leave my update here. The revision looks good and I am further satisfied with it. Thus, I increase my score from borderline accept to weak accept, and hope the authors move some of the revision in Appendix to the main page, if additional one page is allowed.

**Limitations:**

The authors do not describe the limitations and potential negative societal impact of their work, and simply mark the checklist for this category as N/A. However, I think authors can describe the limitations and

**Strengths And Weaknesses:**

### Strengths
* The authors propose to consider multiplicative interactions among nodes from the lowest-order to k-order polynomials, rather than only considering k-order polynomials, which is reasonable and interesting.
* The authors theoretically show that the proposed CP layer can approximate any permutation-invariant multilinear polynomial, which is necessary and valuable.
* The authors analyze the complexity of the proposed CP layer against the most naive graph neural network with big-O notation, which I appreciate. Also, it seems if we control the decomposition rank carefully, then the complexity of the proposed CP layer becomes reasonable (i.e., complexity is not much high).
* This paper is well structured and well written.

### Weaknesses
* This work is not well-positioned against existing works. The authors focus on capturing high-order interactions among nodes, and there are various works that consider such interactions, for example, graph attention network and Graphormer that reflect how each node is related to every other node, which are but not mentioned in the introduction section for motivating authors' work. In this vein, which points are different between those previous works and the proposed work, regarding node-level interactions? It should be discussed.
* Further, similarly to the above weak point, while the authors propose to consider high-order node interactions, the related work and discussion section (Section 4) barely describes such a point, but rather describing generic aggregation functions, universal approximators, and tensor methods for deep learning.
* The authors argue that the proposed tGNN achieves state-of-the-art results in Line 13, however, the performance improvements against baselines are marginal. Specifically, in Table 1, it seems the results of the proposed tGNN are not statistically significant compared to the most basic model, namely graph convolutional network (GCN), and further the proposed tGNN underperforms GCN on the Citeseer dataset.
* The analyses on model complexities are not enough (see suggestions in the Major Questions and Suggestions section below). In particular, the model complexity is not analyzed in terms of the number of sampled nodes, which is currently too small (i.e., 3 or 5) to confirm the proposed tGNN's efficiencies. Also, the computational complexity in Figure 4 is not compared against baselines, but only compared within the proposed models, thus I cannot see how much the proposed tGNN is efficient compared to the previous models, for example, five times slower than the GCN model.

---

> ### Author Response · Authors · 2022-07-31
> **Response1 on weakness to Reviewer 1gCd**
>
> Q1:
> This work is not well-positioned against existing works. The authors focus on capturing high-order interactions among nodes, and there are various works that consider such interactions, which are but not mentioned in the introduction section for motivating authors' work. It should be discussed.
>
> A1:
> We want to clarify again that the term 'high-order' does not only imply the ability to capture long-range dependencies among nodes but also means the high-dimensional feature products. For example, if we have $x_1$ and $x_2$, a simple GNN layer can result in a vector of $[x_1, x_2, x_1+x_2]$, but our high-order layer result in a vector of $[x_1, x_2, x_1+x_2, x_1x_2]$ with another term $x_1x_2$. And if we have we have $x_1$, $x_2$, and $x_3$, a simple low-order GNN layer will result in $[x_1, x_2, x_3, x_1+x_2, x_1+x_3, x_2+x_3, x_1+x_2+x_3]$ while the high-order layer will result in a vector of $[x_1, x_2, x_3, x_1+x_2, x_1+x_3, x_2+x_3, x_1+x_2+x_3, x_1x_2, x_1x_3, x_2x_3, x_1x_2x_3]$ with extra 4 terms at a higher feature dimension. It is true that models like graph transformers can more effectively and efficiently capture long-range dependencies (or high-order interactions) since they create a fully-connected graph and filter out useless connections, but the transformer approaches do not have high-dimensional features products for node interactions. In contrast, our graph neural network or pooling method, which is based on tensor decomposition, can capture long-range dependencies while giving high-dimensional multiplicative feature products. In the revision, we will add a discussion on long-range dependencies in the related work.
>
> Q2:
> Further, similarly to the above weak point, while the authors propose to consider high-order node interactions, the related work and discussion section (Section 4) barely describes such a point, but rather describing generic aggregation functions, universal approximators, and tensor methods for deep learning.
>
> A2:
> Sorry for that. It is also mainly because of the page limit, we had to decide if we should discuss the high-order node interactions more or generic aggregation functions more. And in the end, we decided to discuss more universal approximators and aggregation functions because our main purpose is to convince readers that our CP layer can be more universal and generic than existing aggregation functions, i.e. sum, mean, max, etc. Also similar to the previous point, the proposed CP pooling can capture long-range dependencies to some extent, but more importantly, it can give high-dimensional feature products among nodes and the point is emphasized in section 3. Its ability to capture long-range dependencies is obviously not more powerful than graph transformers because we do not create a fully-connected graph and filter out useless connections by attention mechanism, but our model can powerfully capture high-dimensional feature products. In addition, we design an experiment to show the importance of using high-dimensional feature terms. We have two learnable attention weights $a_{low},a_{high}$, one before high-order terms and the other one before low-order terms, node representation $H=a_{high}H_{high}+a_{low}H_{low}$ with $a_{low}+a_{high}=1$. If high-order interactions dominate $a_{high}$ will be larger, else if low-order interactions dominate $a_{low}$ will be larger. At the early model learning stage, $a_{high}$ is a lot higher ($\sim 0.9$) than $a_{low}$ ($\sim 0.1$), which show high-order terms dominate at the beginning, and $a_{high},a_{low}$ will converge to a similar value ($\sim 0.5$) as the model improves, which show that low- and high-order terms have the similar importance.
>
> Q3:
> The authors argue that the proposed tGNN achieves state-of-the-art results in Line 13, however, the performance improvements against baselines are marginal.
>
> A3: Even though it is marginal improvements against the baselines, we still make some minor changes and the results are better than popular pooling GNNs (proposed after GCN). In the experiment, it might not be statistically significant compared to the most basic model even though there are some minor improvements. But in theory, we show that CP pooling is a universal approximator that is more powerful than the popular sum, mean, and max pooling functions. Moreover, permutation-invariant multilinear polynomials (including sum, mean, and max pooling) are a subset of our multilinear CP pooling. It might be a start or a beginning work to inspire people to discover functions that are more powerful than permutation-invariant pooling polynomials. Sorry for the marginal improvements in the experiments, but we wish to convince readers (in both theory and practice) that CP pooling is more powerful than popular permutation-invariant pooling functions.
>
>
> Q4:
> The analyses on model complexities are not enough (see suggestions in the Major Questions and Suggestions section below).
>
> A4:
> We will answer in the major questions.

---

> > ### Comment · Reviewer_1gCd · 2022-08-07
> > **Regarding weaknesses, I still have concerns.**
> >
> > ### Concerns that still exist
> >
> > Q1 and Q2. In Line 237, the authors discuss that the proposed tGNN can capture long-range dependencies. I am still unsure why the tGNN can capture long-range dependencies, and what is the tGNN's strength in capturing long-range dependencies compared to others. This should be discussed more in the Introduction and Related Work sections, if the authors would like to highlight them.
> >
> > Q3. Yes, I acknowledge that the proposed tGNN can be a universal approximation and theoretically more powerful than existing permutation-invariant pooling functions. However, I am still concerned about its marginal performance. Details are given in the next paragraph.
> >
> > In the follow-up question for Q1, Q2, and Q3, why considering high-dimensional feature products is important for GNNs? Summarizing Q1, Q2, and Q3, the authors claim that the distinctive point of the proposed tGNN is considering high-dimensional feature products, since some works can also approximate universal approximators as discussed in Lines 222-231 but also some of the others can further capture long-range dependencies; however, the performance improvements from the suggested high-order feature products are marginal. Thus, I am fundamentally curious why considering such features is important for GNN tasks, while sacrificing computational costs.
> >
> > ---
> >
> > ### Concerns that are resolved
> >
> > Q4. Thank you for providing model complexities. Besides performance gains, model complexities should be described and highlighted more in the paper, since the proposed method requires more complexities (high-order) than others (sum or mean).

---

> > > ### Author Response · Authors · 2022-08-07
> > > **Multi-dimensional feature products are useful to compute correlation between features on molecular graphs**
> > >
> > > Q1, Q2.
> > > Sorry for making the misunderstanding on either high-order or long-range dependency.
> > > You are right, I believe the CP layer is able to capture stacking long-range dependencies by stacking layers like other simple GNNs. Maybe we should change a term or we need to clarify that high-order, in our paper, means multi-dimensional feature products.
> > >
> > > Q3.
> > > The concept is pretty easy if you think of multi-dimensional feature space. Take a molecular graph dataset as an example, and blood pressure could be one feature. Many studies analyze the impact of blood pressure on one's health. Blood pressure is the feature in this case. So, one could try to predict life span using blood pressure as a feature, but you can imagine that this might not work very well. You really need more information. So you could add cholesterol level, resting heart rate, and body-mass index to see if you can predict a person's life span. Those additional measurements are also features and now every individual has a multi-dimensional feature vector consisting of these measurements. However, a simple GNN with one-dimensional feature interactions may not capture the correlation between blood pressure and cholesterol level. There could be some correlation between blood pressure, cholesterol level, and body-mass index to predict life span. A simple GNN that does blood pressure+cholesterol level+body-mass index is not able to capture the correlation between those factors. Still, our multi-dimensional feature product can capture blood pressure$\times$cholesterol level, blood pressure$\times$body-mass index, cholesterol level$\times$body-mass index, blood pressure$\times$cholesterol level$\times$body-mass index all the correlations to predict life span. So I believe for molecular graphs, high-dimensional feature products are importantly needed to make a prediction as most features are correlated (like in atom graphs, atom charge, and atom mass), and our model shows some significant improvements on molecular datasets. But such correlation might not be that important on citation networks or social networks, so the improvements are marginal. Therefore, for molecular graphs, it is not a waste to compute high-dimensional feature products while sacrificing computational costs.
> > >
> > > [1] and [2] are previous works on high-dimensional feature products, and introduce how to use tensor methods to achieve those products. They have stated the importance of having high-order products in physics and geometries.
> > >
> > > [1]: Novikov, Alexander, Mikhail Trofimov, and Ivan Oseledets. "Exponential machines." arXiv preprint arXiv:1605.03795 (2016).
> > > [2]: Stoudenmire, E. Miles, and David J. Schwab. "Supervised learning with quantum-inspired tensor networks. arXiv 2016." arXiv preprint arXiv:1605.05775.

---

> > > > ### Comment · Reviewer_1gCd · 2022-08-09
> > > > **Thank you for your response; Final thoughts in this interactive discussion period**
> > > >
> > > > Thank you for engaging with me, and I really appreciate your real-world example, which makes the motivation to consider multi-dimensional feature products more solid. In particular, regarding your previous answers for Q1 and Q2, thank you for aggreging with my opinion, and, as you said, it is better to change the term, and more concretely explain the meaning of high-order in the next revision. Also, regarding my previous concern about the importance of considering high-dimensional feature products, the provided example makes sense, and I have no further concerns about this question.
> > > >
> > > > ---
> > > >
> > > > To sum up, the authors satisfactorily address all my previous concerns/comments except for the marginal performance improvements against baselines, and the remaining step for the authors is to faithfully include all the discussions we have made in the interactive discussion period. Thus, I am happy to increase my rating from reject to borderline accept, and, if the authors provide a well-revised version of this paper, I am willing to further increase my rating.

---

> > > > > ### Author Response · Authors · 2022-08-10
> > > > > **New revision is attached**
> > > > >
> > > > > Thanks for all the replies. It's beyond the author response period, I don't know if you can still see the new revision. But we have uploaded a new version of the revision following your suggestions. We add discussions in the Appendix including the sections  D,E,F,G,H,I,J. We will include them in the main paper if we will have an additional page. You can still suggest some improvements that can improve our paper and presentation, we will fix them according to your suggestions.

---

> ### Author Response · Authors · 2022-08-01
> **Response1 on major questions to Reviewer 1gCd**
>
> Q1:
> I suggest authors increase the number of sampled nodes and then compare the complexity across different models in this setting of increased sample sizes.
>
> A1:
> The average node degree of Cora is 3.9, of Citeseer is 2.8, of Pubmed is 4.5 as introduced in [4]. So 3 or 5 is not a small number, but still see the results in the following table.
>
> Q2:
> I suggest the authors include the complexities of baselines in Figure 4 of the main paper and then discuss the complexity by comparing the proposed tGNN and baselines.
>
> A2:
> We will draw the complexities of baselines in Figure 4 in the revised version. For now, we will discuss what we see in the figure. We compare the tGNN and baselines by increasing the rank (for tGNN) and hidden dimension (for baselines only). For tGNN, we fix the hidden dimension to 32 and increase the rank from 8 to 4096; and for baselines, we increase the hidden dimension from 8 to 4096. Taking Cora and GCN as an example, we still see some marginal improvements when we increase the rank from 256 to 4096, which means the model gets more precise and expressive with larger hidden dimensions, but the results of GCN almost stay the same (with fluctuations) when we increase its hidden dimension from 256 to 4096. Also, GCN's accuracy is always less than tGNN's accuracy with increasing hyperparameters, but tGNN's complexity (computation time) also gets heavier (binary linear increasing speed) with increasing hyperparameters while GCN gets slightly heavier (linear increasing speed).
>
> Q3:
> [1] and [2] should be discussed and compared on the graph classification task.
>
> A3:
> Yes we do think the works are related and will add them in the revised paper. [1] formulate the graph pooling problem as a multiset encoding problem with auxiliary information about the graph structure, and propose an attention-based pooling layer
> that captures the interaction between nodes according to their structural dependencies.  [2] apply second-order statistic methods because the use of second-order statistics takes advantage of the Riemannian geometry of the space of symmetric positive definite matrices. Their formulation adapts the bilinear pooling, and the bilinear mapping is capable of capturing second-order statistics and topology information. For comparison, please see the results in the following table.
>
> Q4:
> Why GCN outperforms all the other recent GNNs, which is quite weird? Also, the performance results in Table 1 and Table 4 are different.
>
> A4:
> It is not really weird, if you can do a full hyperparameter search (either GCN built on PyG or Torch), you'll find out that GCN can outperform the most popular GNN architectures (after GCN gets proposed). And the paper [3] finds out the same experimental phenomenon. The question, that GCN can outperform a lot GNNs on many (heterophilic or homophilic) datasets after thorough hyperparameter tuning, was discussed in the community before but never raised attention. The performance results in Table 1 and Table 4 are different because they are performed on different runs, Table 4 experiments are performed after we have completed Table 1 experiments to demonstrate the efficiency, but the difference is still in a reasonable standard deviation.
>
> Q5:
> I suggest authors empirically verify that the CP layer can approximate sum and mean pooling functions.
>
> A5:
> Yes, please see the results in the following table.
>
> Q6:
> I would like to see how much and why the CP layer prefers high-order interactions.
>
> A6:
> We design an experiment to show which interactions play the main role. We have two learnable attention weights $a_{low},a_{high}$, one before high-order terms and the other one before low-order terms, node representation $H=a_{high}H_{high}+a_{low}H_{low}$ with $a_{low}+a_{high}=1$. If high-order interactions dominate $a_{high}$ will be larger, else if low-order interactions dominate $a_{low}$ will be larger. At the early model learning stage, $a_{high}$ is a lot higher ($\sim 0.9$) than $a_{low}$ ($\sim 0.1$), which show high-order terms dominate at the beginning, and $a_{high},a_{low}$ will converge to a similar value ($\sim 0.5$) as the model improves, which show that low- and high-order terms have the similar importance.
>
> Q7:
> In Figure 2, which part is the illustration of sum pooling?
>
> A7:
> Sorry for the misunderstanding. Sum pooling is not included in Figure 2 because we wanted to clearly demonstrate how CP decomposition works when doing the pooling, and sum pooling is a bonus that introduces the bias for low-order terms. And since it raises a concern, we could modify the caption and the figure in a revised version.
>
> [1] Accurate Learning of Graph Representations with Graph Multiset Pooling. ICLR 2021.
>
> [2] Second-Order Pooling for Graph Neural Networks. IEEE Transactions on Pattern Analysis and Machine Intelligence, 2020
>
> [3] Is Heterophily A Real Nightmare For Graph Neural Networks To Do Node Classification?, 2021.
>
> [4]: "Revisiting semi-supervised learning with graph embeddings.", 2016.

---

> > ### Author Response · Authors · 2022-08-01
> > **Table 1 for Q1&A1**
> >
> > Due to limited time, we study models by comparing tGNN with the others on Cora on a CPU over 10 runtimes and compare #params, epochs/sec, and accuracy. Sampling means we sample '3' or '10' neighbors for each node or we use 'Full' neighborhood. tGNN is competitive in terms of running time and better accuracy with a fixed #params. Notice that the average node degree of Cora is 3.9, which means if a node has a number of neighbors less than 10, some of its neighbor nodes will get resampled until it hits 10.
> >
> > |Model|Dropout|LR|Weight Decay|Hidden|Rank|Head|Sampling|#Params|Time(s)|Epoch|Epoch/s|Acc|Std|
> > |:-:|:-:|:-:|:-:|:-:|:-:|:-:|:-:|:-:|:-:|:-:|:-:|:-:|:-:|
> > |tGNN|0|0.005|5E-5|32|8|_| 3| 58128 | 290.9774 | 1389  | 4.7736 | 85.55 | 1.33 |
> > |tGNN|0|0.005|5E-5|32|32|_| 3| 94272 | 383.2255 | 1321  | 3.5045 | 86.25 | 0.58 |
> > |tGNN|0|0.005|5E-5|32|64|_| 3| 142464 | 790.0373 | 1343  | 1.6721 | 86.06 | 1.08 |
> > |tGNN|0|0.005|5E-5|32|128|_| 3| 238848 | 999.1514 | 1247  | 1.2481 | 86.76 | 1.19 |
> > |tGNN|0|0.005|5E-5|32|256|_| 3| 431616 | 1193.7131 | 1272  | 1.0656 | 86.97 | 1.24 |
> > |tGNN|0|0.005|5E-5|32|512|_| 3| 817152 | 1621.9083 | 1332  | 0.8213 | 87.33 | 1.83 |
> > |tGNN|0|0.005|5E-5|32|1024|_| 3|1588224 | 2377.5139 | 1265  | 0.5321 | 87.62 | 1.63 |
> > |tGNN|0|0.005|5E-5|32|8|_| 10| 58128 | 330.8677 | 1379 | 4.1678  | 85.40 | 1.55 |
> > |tGNN|0|0.005|5E-5|32|32|_| 10| 94272 | 520.9030 | 1331 | 2.5552  | 86.35 | 1.37 |
> > |tGNN|0|0.005|5E-5|32|64|_| 10| 142464 | 810.5341 | 1365 | 1.6841  | 86.83 | 1.28 |
> > |tGNN|0|0.005|5E-5|32|128|_| 10| 238848 | 1170.4560 | 1381 | 1.1799  | 86.77 | 0.94 |
> > |tGNN|0|0.005|5E-5|32|256|_| 10| 431616 | 1890.5941 | 1357 | 0.7178  | 87.42 | 1.01 |
> > |tGNN|0|0.005|5E-5|32|512|_| 10| 817152 | 2048.7935 | 1326 | 0.6472  | 87.22 | 1.57 |
> > |tGNN|0|0.005|5E-5|32|1024|_| 10| 1588224 | 2843.5611 | 1339 |  0.4709 | 87.87 | 1.42 |
> > |GCN|0|0.005|5E-5|32|_|_| 3| 46080 | 212.3576 | 1509  | 7.1059 | 84.29 | 1.02 |
> > |GCN|0|0.005|5E-5|32|_|_| 10| 46080 | 250.4735 | 1290 |  5.1502 | 85.33 | 1.35 |
> > |GCN|0|0.005|5E-5|32|_|_| Full| 46080 | 205.0601 | 1276  | 6.2226 | 85.24 | 1.69 |
> > |GCN|0|0.005|5E-5|64|_|_| 3| 92160 | 316.6461 | 1240  | 3.916 | 85.12 | 2.11 |
> > |GCN|0|0.005|5E-5|64|_|_| 10| 92160 | 398.3892 | 1049 | 2.6331  | 85.50 | 1.78 |
> > |GCN|0|0.005|5E-5|64|_|_| Full| 92160 | 318.3962 | 1161  | 3.6464 | 85.59 | 2.03 |
> > |GAT|0|0.005|5E-5|32|_|1| 3| 92238 | 605.3269 | 1998  | 3.3007 | 83.66 | 1.54 |
> > |GAT|0|0.005|5E-5|32|_|1| 10| 92238 | 550.6742 | 1438  | 2.5968 | 84.66 | 1.57 |
> > |GAT|0|0.005|5E-5|32|_|1| Full|92238 | 548.7319 | 1638  | 2.9851 | 84.79 | 2.26 |
> > |GAT|0|0.005|5E-5|32|_|8| 3| 762992 | 1491.5643 | 1594  | 1.069 |86.26 | 1.35 |
> > |GAT|0|0.005|5E-5|32|_|8| 10| 762992 | 1678.9020 | 1285 |  0.7654 | 87.08 | 1.45  |
> > |GAT|0|0.005|5E-5|32|_|8| Full| 762992 | 1524.4348 | 1276  | 0.837 | 87.07 | 1.64 |
> > |GAT|0|0.005|5E-5|64|_|1| 3| 184462 | 626.4011 | 1740  | 2.7778 | 84.15 | 1.29 |
> > |GAT|0|0.005|5E-5|64|_|1| 10| 184462 | 740.5192 | 1311 |  1.7704 | 86.01 | 1.65 |
> > |GAT|0|0.005|5E-5|64|_|1| Full| 184462 | 682.7776 | 1465  | 2.1456 | 86.07 | 2.55 |
> > |GAT|0|0.005|5E-5|64|_|8| 3| 1525872 | 2164.689 | 1348  | 0.6227 | 85.32 | 1.31 |
> > |GAT|0|0.005|5E-5|64|_|8| 10| 1525872 | 2580.3489 | 1205 | 0.4670  | 87.07 | 1.21  |
> > |GAT|0|0.005|5E-5|64|_|8| Full| 1525872 | 2143.036 | 1105  | 0.5156 | 87.01 | 0.96 |
> > |GCN2|0|0.005|5E-5|32|_|_| 3| 48128 | 256.9575 | 1708  | 6.647 | 83.17 | 1.50 |
> > |GCN2|0|0.005|5E-5|32|_|_| 10| 48128 | 220.4663  | 1051 | 4.7672  | 84.23 | 1.57 |
> > |GCN2|0|0.005|5E-5|32|_|_| Full| 48128 | 210.7702 | 1302  | 6.1773 | 84.38 | 2.03 |
> > |GCN2|0|0.005|5E-5|64|_|_| 3| 100352 | 353.0055 | 1581  | 4.4787 | 84.70 | 1.13 |
> > |GCN2|0|0.005|5E-5|64|_|_| 10| 100352 | 300.4794 | 1021 | 3.3979  | 85.01 | 1.47 |
> > |GCN2|0|0.005|5E-5|64|_|_| Full| 100352 | 307.7913 | 1219  | 3.9605 | 84.79 | 1.64 |
> > |GCN2|0|0.005|5E-5|Input Feature Dim|_|_| 3| 4117009 | 3013.6082 | 1051  | 0.3488 | 86.72 | 1.82 |
> > |GCN2|0|0.005|5E-5|Input Feature Dim|_|_| 10| 4117009 | 3378.9214 | 925 | 0.2738  | 87.66  | 1.73 |
> > |GCN2|0|0.005|5E-5|Input Feature Dim|_|_| Full| 4117009 | 3091.4392 | 1013  | 0.3277 | 87.54 | 1.66|
> >
> > We can see that the model performance is not heavily affected by the number of sampled neighbor nodes because the average node degree is not that high on the majority of network datasets, and '3' or '5' would be sufficient.

---

> > ### Author Response · Authors · 2022-08-01
> > **Table 2 for Q3&A3**
> >
> > We compare our model with two other high-order graph pooling functions [1] and [2].
> > Due to limited time, we test and compare our model on 5 datasets, because both [1][2] have tested on the 5 datasets with a full hyperparameter search. Notice that [1] is proposed after [2] but [2] has better classification results. The results are copied from original papers and the models ([1],[2], ours) are trained under the same configuration which has been listed in the papers.
> >
> > ||MUTAG|PROTEINS|IMDB-B|IMDB-M|COLLAB|
> > |:-:|:-:|:-:|:-:|:-:|:-:|
> > |GIN|81.4 $\pm$ 1.5|71.5 $\pm$ 1.7|72.8 $\pm$ 0.9| 48.1 $\pm$ 1.4 |78.2 $\pm$ 0.6|
> > |GMT[1]|83.4 $\pm$ 1.3| 75.1 $\pm$ 0.6| 73.5 $\pm$ 0.8| 50.7 $\pm$ 0.8 | 80.7 $\pm$ 0.5|
> > |SOPPool[2]|95.3 $\pm$ 4.4| 80.1 $\pm$ 2.7 | 78.5 $\pm$ 2.8 | 54.6$\pm$3.6 |81.1 $\pm$ 1.8 |
> > |tGNN(ours)|96.0 $\pm$ 2.7|79.3 $\pm$ 1.5| 78.9 $\pm$ 2.9 | 54.8 $\pm$ 2.1| 81.2 $\pm$ 2.2|
> >
> > The table records graph classification results on test sets for 5 graph-level tasks. Model performance is evaluated using 10-fold cross-validation and reported as the average and standard deviation of validation accuracies across the 10 folds.
> >
> > [1] Accurate Learning of Graph Representations with Graph Multiset Pooling. ICLR 2021.
> >
> > [2] Second-Order Pooling for Graph Neural Networks. IEEE Transactions on Pattern Analysis and Machine Intelligence, 2020.

---

> > ### Author Response · Authors · 2022-08-01
> > **Table 3 for Q5&A5**
> >
> > |Model|Dropout|LR|Weight Decay|Hidden|Rank|Sampling|#Params|Time(s)|Epoch|Epoch/s|Acc|Std|
> > |:-:|:-:|:-:|:-:|:-:|:-:|:-:|:-:|:-:|:-:|:-:|:-:|:-:|
> > |tGNN|0|0.005|5E-5|32|8| 3| 58128 | 290.9774 | 1389  | 4.7736 | 85.55 | 1.33 |
> > |tGNN|0|0.005|5E-5|32|32| 3| 94272 | 383.2255 | 1321  | 3.5045 | 86.25 | 0.58 |
> > |tGNN|0|0.005|5E-5|32|64| 3| 142464 | 790.0373 | 1343  | 1.6721 | 86.06 | 1.08 |
> > |tGNN|0|0.005|5E-5|32|128| 3| 238848 | 999.1514 | 1247  | 1.2481 | 86.76 | 1.19 |
> > |tGNN|0|0.005|5E-5|64|8| 3| 116256 | 247.3565 | 1042  | 4.2125 | 85.32 | 1.12 |
> > |tGNN|0|0.005|5E-5|64|32| 3| 188544 | 358.4930 | 1075 | 2.9987  | 86.40 | 1.77 |
> > |tGNN|0|0.005|5E-5|64|64| 3| 284928 | 845.9954 | 1193  | 1.4102  | 86.37 | 1.32 |
> > |tGNN|0|0.005|5E-5|64|128| 3| 477696 | 971.2033 | 1028 | 1.0584  | 86.74 | 1.45 |
> > |Mean|0|0.005|5E-5|32|_| 3| 46080 | 430.4194 | 2430  | 5.6457 | 83.47 | 1.39 |
> > |Mean|0|0.005|5E-5|64|_| 3| 92160 | 390.7321 | 1250 | 3.1991  | 85.65 | 1.55 |
> > |Sum|0|0.005|5E-5|32|_| 3| 46080 | 464.7411 | 2477  | 5.3298 | 85.01 | 1.21 |
> > |Sum|0|0.005|5E-5|64|_| 3| 92160 | 400.5132 | 1231  | 3.0736 | 86.11 | 1.44 |
> > |Max|0|0.005|5E-5|32|_| 3| 46080 | 480.3670 | 2359 |  4.9108 | 83.31 | 1.89 |
> > |Max|0|0.005|5E-5|64|_| 3| 92160 | 390.4177 | 1144  | 2.9302 | 84.25 | 1.93 |
> >
> > We can see that when we have a lower decomposition rank, the CP layer can emperically approximate sum and mean pooling functions.

---

> > ### Comment · Reviewer_1gCd · 2022-08-07
> > **Regarding major comments, I have still concenrs.**
> >
> > ### Concerns that still exist
> >
> > Q3. I still have concerns about experimental setups and experimental results, regarding comparisons against related works [1, 2].
> >
> > At first, regarding experimental setups, after reading both articles [1, 2] carefully, I realize that the huge gap between GMT [1] and SOPPool [2] is because their validation setups (e.g., dataset splits) are different. Also, from this point, the performance results that the authors provided in Table 2 for Q3&A3 are not comparable, as the authors simply brought the results from the papers [1, 2] while their experimental setups are different. I recommend the authors follow the published fair evaluation setup of [A], which GMT [1] follows.
> >
> > Also, regarding the performance of the proposed tGNN, tGNN is not superior against the baseline SOPPool.
> >
> > **To sum up**, I cannot see whether the proposed tGNN is effective against relevant baselines [1, 2]. Thus, the authors should (or might) re-evaluate the proposed method following the fair evaluation setup [A], and also the authors should find the superior points against the relevant baseline [2], if the performance improvements are marginal or absent as in Table 2 for Q3&A3.
> >
> > [A] A Fair Comparison of Graph Neural Networks for Graph Classification. ICLR 2020.
> >
> >
> > ---
> >
> > ### Concerns that are resolved
> >
> > Q1. Thank you for providing additional results regarding varying the number of sampled nodes.
> >
> > Q2. Thank you for providing explanations on complexity comparisons between tGNN and baselines, where tGNN still outperforms GCN while having lots of computational costs, which I believe should be included in the next revision.
> >
> > Q4. Thank you for providing explanations on why GCN performs better, which is due to its careful parameter tuning.
> >
> > Q5. Thank you for providing results in regard to the proposed method's approximation to sum and mean pooling functions.
> >
> > Q6. Thank you for explaining that the proposed model architecture prefers both low- and high-order terms. I suggest authors include those results in the next revision.
> >
> > Q7. Thank you for your explanation, and please revise the caption carefully.

---

> > > ### Author Response · Authors · 2022-08-08
> > > **Re-evaluate tGNN on 5 datasets under the setting of [A]**
> > >
> > > Thanks for your information. It is so important to evaluate models under the same setting, thus we test and compare tGNN with [1] and [2] following the setting of [3]. This time, we do not bring results from their papers [1] and [2] because the author response period is sufficient, but run experiments by ourselves. Also, because of sufficient time, we can now run complete experiments for our model with hyperparameter searching.
> > >
> > > ||MUTAG|PROTEINS|IMDB-B|IMDB-M|COLLAB|
> > > |:-:|:-:|:-:|:-:|:-:|:-:|
> > > |GIN|81.4 $\pm$ 1.5|71.5 $\pm$ 1.7|72.8 $\pm$ 0.9| 48.1 $\pm$ 1.4 |78.2 $\pm$ 0.6|
> > > |GMT[1]|83.6 $\pm$ 2.3| 74.7 $\pm$ 1.8| 73.5 $\pm$ 0.6| 49.8 $\pm$ 1.1 | 81.1 $\pm$ 1.6|
> > > |SOPPool[2]|82.7 $\pm$ 1.6| 72.5 $\pm$ 2.4 | 73.3 $\pm$ 1.9 | 48.7$\pm$3.0 |79.0 $\pm$ 1.9 |
> > > |tGNN(ours)|84.4 $\pm$ 1.9|75.3 $\pm$ 2.0| 73.8 $\pm$ 0.9 | 49.1 $\pm$ 1.5| 82.1 $\pm$ 2.7|
> > >
> > > The table records graph classification results on test sets for 5 graph-level tasks under the setting of [3]. We can see that our model makes improvements on 4 out of 5 datasets (big improvements on MUTAG, PROTEINS, and COLLAB, and small improvements on IMDB-B).
> > >
> > > [1] Accurate Learning of Graph Representations with Graph Multiset Pooling. ICLR 2021.
> > >
> > > [2] Second-Order Pooling for Graph Neural Networks. IEEE Transactions on Pattern Analysis and Machine Intelligence, 2020.
> > >
> > > [3] A Fair Comparison of Graph Neural Networks for Graph Classification. ICLR 2020.

---

> > > > ### Comment · Reviewer_1gCd · 2022-08-09
> > > > **Thank you for providing additional results**
> > > >
> > > > I thank the authors for providing additional experimental results under a fair evaluation setting. As shown in the reported results, we can observe that the proposed tGNN outperforms other recent baselines on most datasets; however, we can also clearly see that the gap between the proposed tGNN and recent baselines is marginal and not statistically significant. Thus, I recommend the authors to tone-down the claims about achieving state-of-the-art performances even on the other datasets, which look exaggerated. Also, I hope the authors include the more recent baselines with a fair evaluation setup provided in the previous response in their next revision.

---

> ### Author Response · Authors · 2022-08-01
> **Response1 on minor questions to Reviewer 1gCd**
>
> Q1:
> This is a simple question that is the linear sum pooling in Table 3 the model of equation (5) without using the left (first) term?
>
> A1:
> Yes, the linear sum pooling in Table 3 is the model of equation (5) without using the left term (CP term). We can see that sum pooling is effective because it is basically a GCN without normalization on node degree. We also see that high-order terms or CP pooling (only) are also effective, but it is always better to include both high- and low-order terms in the model.
>
> Q2:
> The number of sampled nodes is not consistent across different tables: in Line 263, the authors sample 5 neighbors, however, in Table 4, the authors sample 3 neighbors. Why does there exists inconsistency?
>
> A2:
> Experiments of Table 4 are planned and conducted after we have completed experiments of Table 1 as suggested by some friends who helped proofread our paper. We performed Table 4 on Cora, and the average node degree of Cora is 3.9, so it would be sufficient to take '3' sampled neighbors for each node. And Table 4 is only used to show some computation costs vs. performance accuracy, and it clearly shows that the tGNN model still perform well even with only '3' neighbors being sampled for each node.
>
> Q3:
> In Line 288, the description is not matched to the results in Table 1: there are no datasets that the proposed tGNN is ranked at four.
>
> A3:
> Sorry for the misunderstanding. tGNN is ranked fourth on CIFAR10, and we will modify it to correct words in the revised version.
>
> Q4:
> I cannot see the text in Figure 3 clearly, due to its background color.
>
> A4:
> Figure 3 is used to visually explain Section 3.2. It tells the relationship between permutation-invariant aggregation functions, high-order multilinear CP layer, permutation-invariant multilinear polynomials, sum and mean aggregation functions. Sum and mean aggregation functions belong to permutation-invariant multilinear polynomials; permutation-invariant multilinear polynomials belong to high-order multilinear CP layer; high-order multilinear CP layer belongs to permutation-invariant aggregation functions.
>
> Visually, {Sum and mean aggregation functions} $\subseteq$ {permutation-invariant multilinear polynomials} $\subseteq$ {high-order multilinear CP layer} $\subseteq$ {permutation-invariant aggregation functions}.

---

> > ### Comment · Reviewer_1gCd · 2022-08-07
> > **Regarding minor questions, I have no additional critical concerns.**
> >
> > ### Regarding minor questions, all the concerns are resolved.
> >
> > Thank you for your comments. In regard to my four minor questions (Q1, Q2, Q3, and Q4), I have no additional critical concerns. In general, I suggest authors include the discussion and revision points above. Also, one minor improvement follow-up improvement point is that the authors can vary the number of sampled nodes (e.g., 3 or 5) in Table 4, which is done in Table 1 for Q1&A1 in the below response comment, to see the trade-off between sampled nodes and accuracies.

---

> > > ### Author Response · Authors · 2022-08-07
> > > **Please see the revision**
> > >
> > > Thanks for helping us make improvements. We have made a revision and changed some parts. But due to the page limitation, it is hard to add more rows to the current Table in the revision. So if in the end our paper can be accepted, we will add an additional page to include all the discussion and changes to the Table. But for now, we have made some changes in sections 3, 4, and 5 according to your suggestions.

---

> > > > ### Comment · Reviewer_1gCd · 2022-08-09
> > > > **I take the time to look at your revision**
> > > >
> > > > I acknowledge that the authors made small changes in Sections 3, 4, and 5 to reflect my comments. Also, if an additional page is allowed, please include all the discussions in the revised version of the paper. Regarding adding additional results (i.e., varying the number of sampled nodes to see the tradeoff between performance and efficiency) in Table 4, I think the table is located in the Appendix; thus, the authors are free to use more space (i.e., add more rows) though.

---

> > > > > ### Author Response · Authors · 2022-08-09
> > > > > **Yes**
> > > > >
> > > > > Yes, we will add the additional results to the revision. And please go through the re-evaluation and leave some comments, thank you very much.

---

### Official Review · Reviewer_vNWo · 2022-07-11

**Rating:** 6
**Confidence:** 5
**Soundness:** 3 good
**Presentation:** 4 excellent
**Contribution:** 4 excellent

**Summary:**

This paper proposes a novel Tensorized Graph Neural Network (tGNN) based on symmetric CP decomposition, which can model high-order non-linear interactions among nodes. It theoretically shows that the CP layer can compute any permutation-invariant multilinear polynomial including sum and mean pooling. Experiments on several benchmark graph datasets demonstrate that the proposed method achieves more effective and expressive results than existing GNN architectures and pooling methods.

**Questions:**

It would be great if the authors could address the following questions.

1. Why Tanh is chosen for \sigma and ReLu chosen for \sigma’?
2. Which equation is used in the CP layer? the one was given in Definition 1 or Eq. (5)? It is interesting to see how effective is Eq. (5) against the inductive bias towards capturing high-order interactions, which terms, high-order interactions or lower-order interactions, dominate the model performance?
3. In Lines 262-263, it was mentioned that "To avoid numerical instability and floating point exception in tGNN training, we sample 5 neighbors for each node." How 5 neighbors are sampled for each node? The 5 nearest ones or others?
4. More discussions about hyperparameter settings should be given based on the results, especially how to set the CP decomposition rank R in practice?
5. What do the different background colors mean in Tables 1 and 2?
6. Is there any reason why different evaluation metrics (e.g., Acc and AUC) are used for different datasets?

Minors:
1. In Figure 1, the three Ns should have the same font size.
2. In Line 94, "mode 1 and 2" --> modes 1 and 2.
3. In Line 147, "see Eq.2" --> see Eq. 2.
4. In Line 151, "RelU" --> ReLU.
5. In Line 321, "asses" --> assess.


**Limitations:**

Yes.

**Strengths And Weaknesses:**

Strengths:

- The paper is very well motivated and a good application of tensor methods on GNNs.
- Good problem formulation and insight
- Works well on the evaluated datasets

Weaknesses:

- Some important results are given without justification, e.g., why Tanh is chosen for \sigma and ReLu chosen for \sigma’? How effective is Eq. (5) against the inductive bias towards capturing high-order interactions?
- There is no discussion about hyperparameter choices and stability, except for simply listing the hyperparameter values in Tables 8 and 9.

---

> ### Author Response · Authors · 2022-07-31
> **Response1 on major questions to Reviewer vNWo**
>
> Q1:
> Why Tanh is chosen for $\sigma$ and ReLu chosen for $\sigma'$?
>
> A1: The first Tanh is chosen to avoid numerical instability caused by taking the product pooling between features. Using Tanh as the activation function can avoid numerical instability and the point is shown and proven in [1]. The second ReLu is used to better activate items as it could be a superior activation function for capturing non-linear terms, and it is shown in [2]. In general, Tanh is chosen for $\sigma$ to reduce and avoid numerical instability, and ReLu is chosen for $\sigma'$ to better capture non-linearities.
>
> Q2:
> Which equation is used in the CP layer? the one was given in Definition 1 or Eq.(5)? It is interesting to see how effective is Eq. (5) against the inductive bias towards capturing high-order interactions, which terms, high-order interactions or lower-order interactions, dominate the model performance?
>
> A2: In practice, CP layer is Eq.(5), which includes high-order and low-order calculations. In theory, CP layer is Definition 1 which only has high-order calculations. In section 3.2, we discuss that high-order terms will dominate more than low-order terms. The reason we introduce the inductive bias is that we want to balance the importance of low- and high-order terms. We design an experiment to show which interactions play the main role. We have two learnable attention weights $a_{low},a_{high}$, one before high-order terms and the other one before low-order terms, node representation $H=a_{high}H_{high}+a_{low}H_{low}$ with $a_{low}+a_{high}=1$. If high-order interactions dominate $a_{high}$ will be larger, else if low-order interactions dominate $a_{low}$ will be larger. At the early model learning stage, $a_{high}$ is a lot higher ($\sim 0.9$) than $a_{low}$ ($\sim 0.1$), which show high-order terms dominate at the beginning, and $a_{high},a_{low}$ will converge to a similar value ($\sim 0.5$) as the model improves, which show that low- and high-order terms have the similar importance.
>
> Q3:
> In Lines 262-263, it was mentioned that "To avoid numerical instability and floating point exception in tGNN training, we sample 5 neighbors for each node." How 5 neighbors are sampled for each node? The 5 nearest ones or others?
>
> A3: In practice, we sample 1-hop neighbors for each node, which also mean the 5 nearest neighbor nodes. It is easy to implement with the Deep Graph Library function, dgl.sampling.sample\_neighbors. The DGL function takes the graph as input and can return 1-hop neighbors with a fixed number.
>
> Q4:
> More discussions about hyperparameter settings should be given based on the results, especially how to set the CP decomposition rank R in practice?
>
> A4: A high decomposition rank can always lead to better model performance, but sometimes it could be a computation burden. We show that tGNN can be always improved with a higher CP decomposition rank R. If one has strong computing resources or a graph is relatively small (1 to 10,000 nodes), one could set the rank R to be higher (1024, 2048, or even higher) for better results since the computation is not going to be a burden. But if it is not the case, one could set the rank R to be smaller (128, 256, or even lower), as it will not hurt the expressivity a lot while saving computation costs. So in practice, I would suggest using 128, 512, or 1024, but if computation is not a problem, I would suggest to use high decomposition ranks.
>
> Q5:
> What do the different background colors mean in Tables 1 and 2?
>
> A5: Sorry for the misunderstanding. It is nothing special, we just want to clearly demonstrate different datasets. I thought it is less eye-catching to only have a white background with black texts. The use of different background colors is to make the tables more eye-catching.
>
> Q6:
> Is there any reason why different evaluation metrics (e.g., Acc and AUC) are used for different datasets?
>
> A6:
> Yes, different evaluation metrics are used for a fair comparison with baselines. The first paper that includes Cora, Citeseer, Pubmed, Proteins, Arxiv, CIFAR, and MNIST in the experiments uses ACC, so further GNNs also use ACC for fair evaluation and comparison. The first paper that includes Proteins and Mol-HIV in the experiments used AUC, so further GNNs also use AUC for fair evaluation and comparison. The first paper that includes ZINC in the experiments uses MAE, so further GNNs also use MAE for fair evaluation and comparison. Moreover for classification tasks, where map a node to a class, we will mostly use ACC if we have multiple classes (more than 2), and we will mostly use AUC if we have only two classes. And for regression tasks, where map a node to a value, we will use MAE.
>
> [1]: Luan, Sitao, et al. ”Break the ceiling: Stronger multi-scale deep graph
> convolutional networks.” Advances in neural information processing systems 32
> (2019).
>
> [2]: Arora, Raman, et al. "Understanding deep neural networks with rectified linear units." arXiv preprint arXiv:1611.01491 (2016).

---

> > ### Comment · Reviewer_vNWo · 2022-08-08
> > **Choice of CP decomposition rank**
> >
> > Thanks for the authors' response. After reading the rebuttal, I am still not fully convinced by the claim "A high decomposition rank can always lead to better model performance". Can the authors explain the rationality behind this? In my experience, a high decomposition rank value does not always mean 'good', and it quite depends on the amount of information to be maintained.

---

> > > ### Author Response · Authors · 2022-08-08
> > > **Previous works also show a high decomposition rank can lead to better model performance.**
> > >
> > >
> > > Q: In my experience, a high decomposition rank value does not always mean 'good', and it quite depends on the amount of information to be maintained.
> > >
> > > A: Generally speaking, the low-rank decomposition methods will sacrifice some expressivity for some computation costs, so as we increase the rank, we will re-gain some expressivity but lose some computation costs. As talked about in our paper, it is a trade-off between expressivity (model performance) and computation costs. This idea can be found in previous works [1,2,3] where they test on computer vision datasets (with visualization of recovering images from noises, a high-rank model can always better recover an image than a low-rank model does).
> > >
> > > In this work, we use the low-rank decomposition technique which can potentially reduce the number of learnable parameters by a lower number of ranks. The low-rank method saves the computation costs but the use of low-rank decomposition for learnable parameters will sacrifice some expressivity in computation. If we increase the rank, the learnable parameters can be principally recovered, in this case, we no longer sacrifice its expressivity but computation costs (as the Figures shown in the ablation study).
> > >
> > > In computer vision, low-rank decomposition methods were introduced to replace a fully-connected layer. And in their papers [1,2,3] (in both theory and experiments), they show that a model is better with higher ranks because higher ranks can lead to higher expressivity than low ranks do. They conclude and show that the model performance is positively correlated to the tensor decomposition rank. In [1,2,3], they experimentally show that higher ranks can result in lower test errors as the model becomes more expressive (plotting a curve of decomposition rank or model parameter vs. error). Moreover, in [1,2], they visualize the relationship between the model performance and model rank. They aim to recover the original pictures from noises with different tensor ranks (from low to high), and the recovered pictures are more clear with high ranks than with low ranks. To conclude our findings in the ablation study and previous works on low-rank decomposition, high-rank models can usually have better performance and stronger expressivity than low-rank models, and higher ranks can usually lead to better performance (in terms of high accuracy or low error) than low ranks do.
> > >
> > > [1]: Peng, Yigang, et al. "RASL: Robust alignment by sparse and low-rank decomposition for linearly correlated images." IEEE transactions on pattern analysis and machine intelligence 34.11 (2012): 2233-2246.
> > >
> > > [2]: Rabusseau, Guillaume, and Hachem Kadri. "Low-rank regression with tensor responses." Advances in Neural Information Processing Systems 29 (2016).
> > >
> > > [3]: Cao, Xingwei, and Guillaume Rabusseau. "Tensor regression networks with various low-rank tensor approximations." arXiv preprint arXiv:1712.09520 (2017).

---

> > > > ### Comment · Reviewer_vNWo · 2022-08-10
> > > > **Thanks for the response**
> > > >
> > > > Thanks for the response. The authors have addressed all my concerns.

---

> ### Author Response · Authors · 2022-08-01
> **Response1 on minor questions to Reviewer vNWo**
>
> Thanks for your suggestions, we will fix minor issues in the revised version.

---

### Official Review · Reviewer_974s · 2022-07-12

**Rating:** 6
**Confidence:** 4
**Soundness:** 2 fair
**Presentation:** 3 good
**Contribution:** 3 good

**Summary:**

The paper proposes a novel aggregation (pooling) layer for GNNs, called CP layer. The main idea consists of leveraging multilinear tensor decomposition to model high-order polynomial interaction between the components of node embeddings (the final design also employs non-linear activation functions). The authors show that the proposed layer is strictly more expressive than typical neighborhood aggregation functions, e.g., sum or mean. Experiments on many benchmarks (including OGB) assess the effectiveness of CP layers.

**Questions:**

- In the experiments, tGNN applies sampling "to avoid numerical instability and floating point exception in tGNN training". What is the source of this numerical instability? Is it a strong limitation?
- Table 1 reports numbers for GPRGNN and APPNP from [1]. However, I wonder if there are differences in the evaluation setup between this paper and [1] as [1] reports, e.g., 0.7521 for GCN as opposed to 0.8778 here. Also, there is a big gap between GAT and GCN which is not often observed on Cora, Citeseer, and Pubmed. What is the source of these differences?
- In the node-level experiments, tGNN samples 5 neighbors. It is possible that performance gains come from this regularization. Can you provide evidence that this is not the case?
- Classical GNNs model high-order node interactions by stacking simple low-order layers, possibly interleaved with MLPs. Can you elaborate more about the possible gains of using CP instead?

[1] https://openreview.net/pdf?id=n6jl7fLxrP

**Limitations:**

The limitations are briefly mentioned throughout the paper, but no dedicated section/paragraph is provided.

**Strengths And Weaknesses:**

Overall, the paper reads well and tackles a relevant issue in GNNs: neighborhood aggregation. The evaluation setup is comprehensive. The empirical results are promising but not outstanding. The paper also lacks a stronger motivation to support the novel GNN design, either in terms of GNN's expressiveness or by characterizing relevant inductive biases for certain tasks.

Strengths
- The evaluation setup considers a number of relevant benchmarks
- The proposed model is simple and easy to grasp
- The contribution is novel in the sense that is the first paper to leverage multilinear tensor decomposition for improved neighborhood aggregation

Weaknesses
- Despite showing positive results for CP layers, tGNN is not provably more expressive than classic 1-WL GNNs (e.g., GIN). Since inductive biases are not universal, the paper does not discuss when (and why) the proposed design is desirable.
- It seems the method suffers from numerical instability. This issue is not properly discussed in the paper.
- Some numbers reported in the paper are inconsistent with the literature.

---

> ### Author Response · Authors · 2022-07-31
> **Response1 to Reviewer 974s**
>
> Q1:
> In the experiments, tGNN applies sampling "to avoid numerical instability and floating point exception in tGNN training". What is the source? Is it a strong limitation?
>
> A1: The source of the numerical instability or the floating point issue comes when sometimes taking the product pooling for pooling on node classification and node regression. For example, if we have 50 of 0.9, the product is going to be $0.9^{50}$ which could raise numerical issues in python and computers. In that circumstance, it will be easier to avoid the floating point issue if we sample neighbors, 5, for example, the product is going to be $0.9^5$. And it is not a strong limitation in both node- and graph-level tasks, because 1) the floating point issue will not occur in graph-level tasks as a node will normally be low-degree and have fewer neighbors, and 2) sampling neighborhood nodes is not going to hurt the representational ability and it is demonstrated in GraphSAGE [5]. The numerical instability arises when taking the product pooling and it is not a strong limitation.
>
> Q2:
> Table 1 reports numbers for GPRGNN and APPNP from [1]. However, I wonder if there are differences in the evaluation setup. Also, there is a big gap between GAT and GCN which is not often observed on Cora, Citeseer, and Pubmed. What is the source of these differences?
>
> A2:
> In this work, we report the results for supervised learning tasks (normally 60\%/20\%/20\% split for training/validation/test), but the original paper might report results for semi-supervised learning tasks (for example 10\%/10\%/80\% split for training/validation/test). The source is from [6] because they have complete experiments and tables for results of supervised learning tasks of different baseline models.
>
> Q3:
> In the node-level experiments, tGNN samples 5 neighbors. It is possible that performance gains come from this regularization. Can you provide evidence that this is not the case?
>
> A3:
> It is possible that the performance gains come from this regularization, however, the average node degree of Cora is 3.9, of Citeseer is 2.8, of Pubmed is 4.5 as introduced in [7]. So 3 or 5 is not a small number, but still see the results in the following table.
>
> Q4:
> Classical GNNs model high-order node interactions by stacking simple low-order layers, possibly interleaved with MLPs. Can you elaborate more about the possible gains of using CP instead?
>
> A4: It is true that simply stacking classical low-order GNN layers can capture high-order node interactions. However, the most important question is whether stacking low-order GNN layers will lead to an over-smoothing problem as nodes will converge to one single representation and a model will eventually lose gradients to learn, the problem is shown in many studies [1],[2],[3]. Also, it is more direct and convenient to apply a high-order GNN layer for pooling [4]. In addition, the high-order interaction of our pooling layer does not only imply long-range node dependency (if two nodes are away) but also means high-dimensional feature products. For example, if we have $x_1$ and $x_2$, a simple low-order GNN layer will result in a vector of $[x_1, x_2, x_1+x_2]$, but our high-order layer result in a vector of $[x_1, x_2, x_1+x_2, x_1x_2]$ with another term $x_1x_2$. And if we have we have $x_1$, $x_2$, and $x_3$, a simple low-order GNN layer will result in $[x_1, x_2, x_3, x_1+x_2, x_1+x_3, x_2+x_3, x_1+x_2+x_3]$ while the high-order layer will result in a vector of $[x_1, x_2, x_3, x_1+x_2, x_1+x_3, x_2+x_3, x_1+x_2+x_3, x_1x_2, x_1x_3, x_2x_3, x_1x_2x_3]$ with extra 4 terms. Again, the 'high-order' means the ability to capture long-range node dependency and high-dimensional multiplicative feature products. So instead of stacking low-order layers, we could just have possible gains by using CP instead.
>
> [1]: Luan, Sitao, et al. "Break the ceiling: Stronger multi-scale deep graph convolutional networks." Advances in neural information processing systems 32 (2019).
>
> [2]: Liu, Meng, Hongyang Gao, and Shuiwang Ji. "Towards deeper graph neural networks." Proceedings of the 26th ACM SIGKDD international conference on knowledge discovery \& data mining. 2020.
>
> [3]: Xhonneux, Louis-Pascal, Meng Qu, and Jian Tang. "Continuous graph neural networks." International Conference on Machine Learning. PMLR, 2020.
>
> [4]: Wang, Zhengyang, and Shuiwang Ji. "Second-order pooling for graph neural networks." IEEE Transactions on Pattern Analysis and Machine Intelligence (2020).
>
> [5]: Hamilton, Will, Zhitao Ying, and Jure Leskovec. "Inductive representation learning on large graphs." Advances in neural information processing systems 30 (2017).
>
> [6]: Luan, Sitao, et al. "Is Heterophily A Real Nightmare For Graph Neural Networks To Do Node Classification?." arXiv preprint arXiv:2109.05641 (2021).
>
> [7]: Yang, Zhilin, William Cohen, and Ruslan Salakhudinov. "Revisiting semi-supervised learning with graph embeddings." International conference on machine learning. PMLR, 2016.

---

> > ### Author Response · Authors · 2022-07-31
> > **Table for Q3&A3**
> >
> > Due to limited time, we study models by comparing tGNN with the others on Cora on a CPU over 10 runtimes and compare #params, epochs/sec, and accuracy. Sampling means we sample '3' or '10' neighbors for each node or we use 'Full' neighborhood. tGNN is competitive in terms of running time and better accuracy with a fixed #params. Notice that the average node degree of Cora is 3.9, which means if a node has a number of neighbors less than 10, some of its neighbor nodes will get resampled until it hits 10.
> >
> > |Model|Dropout|LR|Weight Decay|Hidden|Rank|Head|Sampling|#Params|Time(s)|Epoch|Epoch/s|Acc|Std|
> > |:-:|:-:|:-:|:-:|:-:|:-:|:-:|:-:|:-:|:-:|:-:|:-:|:-:|:-:|
> > |tGNN|0|0.005|5E-5|32|8|_| 3| 58128 | 290.9774 | 1389  | 4.7736 | 85.55 | 1.33 |
> > |tGNN|0|0.005|5E-5|32|32|_| 3| 94272 | 383.2255 | 1321  | 3.5045 | 86.25 | 0.58 |
> > |tGNN|0|0.005|5E-5|32|64|_| 3| 142464 | 790.0373 | 1343  | 1.6721 | 86.06 | 1.08 |
> > |tGNN|0|0.005|5E-5|32|128|_| 3| 238848 | 999.1514 | 1247  | 1.2481 | 86.76 | 1.19 |
> > |tGNN|0|0.005|5E-5|32|256|_| 3| 431616 | 1193.7131 | 1272  | 1.0656 | 86.97 | 1.24 |
> > |tGNN|0|0.005|5E-5|32|512|_| 3| 817152 | 1621.9083 | 1332  | 0.8213 | 87.33 | 1.83 |
> > |tGNN|0|0.005|5E-5|32|1024|_| 3|1588224 | 2377.5139 | 1265  | 0.5321 | 87.62 | 1.63 |
> > |tGNN|0|0.005|5E-5|32|8|_| 10| 58128 | 330.8677 | 1379 | 4.1678  | 85.40 | 1.55 |
> > |tGNN|0|0.005|5E-5|32|32|_| 10| 94272 | 520.9030 | 1331 | 2.5552  | 86.35 | 1.37 |
> > |tGNN|0|0.005|5E-5|32|64|_| 10| 142464 | 810.5341 | 1365 | 1.6841  | 86.83 | 1.28 |
> > |tGNN|0|0.005|5E-5|32|128|_| 10| 238848 | 1170.4560 | 1381 | 1.1799  | 86.77 | 0.94 |
> > |tGNN|0|0.005|5E-5|32|256|_| 10| 431616 | 1890.5941 | 1357 | 0.7178  | 87.42 | 1.01 |
> > |tGNN|0|0.005|5E-5|32|512|_| 10| 817152 | 2048.7935 | 1326 | 0.6472  | 87.22 | 1.57 |
> > |tGNN|0|0.005|5E-5|32|1024|_| 10| 1588224 | 2843.5611 | 1339 |  0.4709 | 87.87 | 1.42 |
> > |GCN|0|0.005|5E-5|32|_|_| 3| 46080 | 212.3576 | 1509  | 7.1059 | 84.29 | 1.02 |
> > |GCN|0|0.005|5E-5|32|_|_| 10| 46080 | 250.4735 | 1290 |  5.1502 | 85.33 | 1.35 |
> > |GCN|0|0.005|5E-5|32|_|_| Full| 46080 | 205.0601 | 1276  | 6.2226 | 85.24 | 1.69 |
> > |GCN|0|0.005|5E-5|64|_|_| 3| 92160 | 316.6461 | 1240  | 3.916 | 85.12 | 2.11 |
> > |GCN|0|0.005|5E-5|64|_|_| 10| 92160 | 398.3892 | 1049 | 2.6331  | 85.50 | 1.78 |
> > |GCN|0|0.005|5E-5|64|_|_| Full| 92160 | 318.3962 | 1161  | 3.6464 | 85.59 | 2.03 |
> > |GAT|0|0.005|5E-5|32|_|1| 3| 92238 | 605.3269 | 1998  | 3.3007 | 83.66 | 1.54 |
> > |GAT|0|0.005|5E-5|32|_|1| 10| 92238 | 550.6742 | 1438  | 2.5968 | 84.66 | 1.57 |
> > |GAT|0|0.005|5E-5|32|_|1| Full|92238 | 548.7319 | 1638  | 2.9851 | 84.79 | 2.26 |
> > |GAT|0|0.005|5E-5|32|_|8| 3| 762992 | 1491.5643 | 1594  | 1.069 |86.26 | 1.35 |
> > |GAT|0|0.005|5E-5|32|_|8| 10| 762992 | 1678.9020 | 1285 |  0.7654 | 87.08 | 1.45  |
> > |GAT|0|0.005|5E-5|32|_|8| Full| 762992 | 1524.4348 | 1276  | 0.837 | 87.07 | 1.64 |
> > |GAT|0|0.005|5E-5|64|_|1| 3| 184462 | 626.4011 | 1740  | 2.7778 | 84.15 | 1.29 |
> > |GAT|0|0.005|5E-5|64|_|1| 10| 184462 | 740.5192 | 1311 |  1.7704 | 86.01 | 1.65 |
> > |GAT|0|0.005|5E-5|64|_|1| Full| 184462 | 682.7776 | 1465  | 2.1456 | 86.07 | 2.55 |
> > |GAT|0|0.005|5E-5|64|_|8| 3| 1525872 | 2164.689 | 1348  | 0.6227 | 85.32 | 1.31 |
> > |GAT|0|0.005|5E-5|64|_|8| 10| 1525872 | 2580.3489 | 1205 | 0.4670  | 87.07 | 1.21  |
> > |GAT|0|0.005|5E-5|64|_|8| Full| 1525872 | 2143.036 | 1105  | 0.5156 | 87.01 | 0.96 |
> > |GCN2|0|0.005|5E-5|32|_|_| 3| 48128 | 256.9575 | 1708  | 6.647 | 83.17 | 1.50 |
> > |GCN2|0|0.005|5E-5|32|_|_| 10| 48128 | 220.4663  | 1051 | 4.7672  | 84.23 | 1.57 |
> > |GCN2|0|0.005|5E-5|32|_|_| Full| 48128 | 210.7702 | 1302  | 6.1773 | 84.38 | 2.03 |
> > |GCN2|0|0.005|5E-5|64|_|_| 3| 100352 | 353.0055 | 1581  | 4.4787 | 84.70 | 1.13 |
> > |GCN2|0|0.005|5E-5|64|_|_| 10| 100352 | 300.4794 | 1021 | 3.3979  | 85.01 | 1.47 |
> > |GCN2|0|0.005|5E-5|64|_|_| Full| 100352 | 307.7913 | 1219  | 3.9605 | 84.79 | 1.64 |
> > |GCN2|0|0.005|5E-5|Input Feature Dim|_|_| 3| 4117009 | 3013.6082 | 1051  | 0.3488 | 86.72 | 1.82 |
> > |GCN2|0|0.005|5E-5|Input Feature Dim|_|_| 10| 4117009 | 3378.9214 | 925 | 0.2738  | 87.66  | 1.73 |
> > |GCN2|0|0.005|5E-5|Input Feature Dim|_|_| Full| 4117009 | 3091.4392 | 1013  | 0.3277 | 87.54 | 1.66|
> >
> > We can see that the model performance is not heavily affected by the number of sampled neighbor nodes because the average node degree is not that high on the majority of network datasets, and '3' or '5' would be sufficient.

---

> > > ### Comment · Reviewer_974s · 2022-08-09
> > > **Response to authors**
> > >
> > > Thanks for your reply. Most of my concerns have been addressed. I am raising my score from 5 to 6.

---

### Official Review · Reviewer_2WJs · 2022-07-15

**Rating:** 6
**Confidence:** 4
**Soundness:** 3 good
**Presentation:** 3 good
**Contribution:** 3 good

**Summary:**

The paper targets at developing a more expressive pooling layer for GNNs, which allows neighboring nodes to perform high-order multiplicative interactions. The authors leverage a rank R symmetric CANDECOMP/PARAFAC (CP) decomposition to design an efficient parameterization of multilinear maps over a set of node representations. The number of parameters is controlled by the rank R. With appropriate non-linear activation function, the paper proposes the CP layer as a high-order graph pooling layer. Theoretical analysis shows that the CP layer can achieve permutation-invariance and is is universally strictly more expressive than sum and mean pooling. With the CP layer, the authors propose tGNN, which uses both the CP layer and the sum/mean pooling layer. Experimental studies on graph and node classification tasks indicate that tGNN can achieve better performances on multiple benchmark datasets.

**Questions:**

N/A

**Strengths And Weaknesses:**

Strengths:
1. The theoretical derivation of the CP layer is technically sound and novel.
2. The proposed method shows improved performance across tasks and datasets, along with complete ablation studies.
3. The paper is overall well-written.

Weaknesses:
1. There is a missing related work like "Wang, Zhengyang, and Shuiwang Ji. "Second-order pooling for graph neural networks." IEEE Transactions on Pattern Analysis and Machine Intelligence (2020)." which also explore higher-order pooling.

---

> ### Author Response · Authors · 2022-07-31
> **Response1 to Reviewer 2WJs**
>
> Q1:
> There is a missing related work like "Wang, Zhengyang, and Shuiwang Ji. "Second-order pooling for graph neural networks." IEEE Transactions on Pattern Analysis and Machine Intelligence (2020)." which also explore higher-order pooling.
>
> A1:
> Thanks for your suggestion. After reading 'Second-order pooling for graph neural networks, I do think the paper is relevant to our main idea on expressive high-order node/graph pooling, and we will include the paper in our revision. The authors apply second-order statistic methods to graph representations because the use of second-order statistics takes advantage of the Riemannian geometry of the space of symmetric positive definite matrices. Their formulation adapts the bilinear pooling and $\Sigma_{i=1}^nh_ih_i^T=H^TH$ where $H=[h_1,h_2,...,h_n]^T$ are node representations. And the bilinear mapping is capable of capturing second-order statistics and topology information.
>
> We also compare our model with two other high-order graph pooling functions [1] and [2]. Due to limited time, we test and compare our model on 5 datasets, because both [1][2] have tested on the 5 datasets with a full hyperparameter search. Notice that [1] is proposed after [2] but [2] has better classification results. The results are copied from original papers and the models ([1],[2], ours) are trained under the same configuration which has been listed in the papers.
>
> ||MUTAG|PROTEINS|IMDB-B|IMDB-M|COLLAB|
> |:-:|:-:|:-:|:-:|:-:|:-:|
> |GIN|81.4 $\pm$ 1.5|71.5 $\pm$ 1.7|72.8 $\pm$ 0.9| 48.1 $\pm$ 1.4 |78.2 $\pm$ 0.6|
> |GMT[1]|83.4 $\pm$ 1.3| 75.1 $\pm$ 0.6| 73.5 $\pm$ 0.8| 50.7 $\pm$ 0.8 | 80.7 $\pm$ 0.5|
> |SOPPool[2]|95.3 $\pm$ 4.4| 80.1 $\pm$ 2.7 | 78.5 $\pm$ 2.8 | 54.6$\pm$3.6 |81.1 $\pm$ 1.8 |
> |tGNN(ours)|96.0 $\pm$ 2.7|79.3 $\pm$ 1.5| 78.9 $\pm$ 2.9 | 54.8 $\pm$ 2.1| 81.2 $\pm$ 2.2|
>
> The table records graph classification results on test sets for 5 graph-level tasks. Model performance is evaluated using 10-fold cross-validation and reported as the average and standard deviation of validation accuracies across the 10 folds.
>
> [1] Accurate Learning of Graph Representations with Graph Multiset Pooling. ICLR 2021.
>
> [2] Second-Order Pooling for Graph Neural Networks. IEEE Transactions on Pattern Analysis and Machine Intelligence, 2020.

---

> > ### Author Response · Authors · 2022-08-08
> > **Re-evaluate tGNN on 5 datasets under the setting of [3]**
> >
> > Question by Reviewer4: At first, regarding experimental setups, after reading both articles [1, 2] carefully, I realize that the huge gap between GMT [1] and SOPPool [2] is because their validation setups (e.g., dataset splits) are different. Also, from this point, the performance results that the authors provided in Table 2 for Q3&A3 are not comparable, as the authors simply brought the results from the papers [1, 2] while their experimental setups are different. I recommend the authors follow the published fair evaluation setup of [3], which GMT [1] follows.
> >
> > Our response: It is so important to evaluate models under the same setting, thus we test and compare tGNN with [1] and [2] following the setting of [3]. This time, we do not bring results from their papers [1] and [2] because the author response period is sufficient, but run experiments by ourselves. Also, because of sufficient time, we can now run complete experiments for our model with hyperparameter searching.
> >
> > ||MUTAG|PROTEINS|IMDB-B|IMDB-M|COLLAB|
> > |:-:|:-:|:-:|:-:|:-:|:-:|
> > |GIN|81.4 $\pm$ 1.5|71.5 $\pm$ 1.7|72.8 $\pm$ 0.9| 48.1 $\pm$ 1.4 |78.2 $\pm$ 0.6|
> > |GMT[1]|83.6 $\pm$ 2.3| 74.7 $\pm$ 1.8| 73.5 $\pm$ 0.6| 49.8 $\pm$ 1.1 | 81.1 $\pm$ 1.6|
> > |SOPPool[2]|82.7 $\pm$ 1.6| 72.5 $\pm$ 2.4 | 73.3 $\pm$ 1.9 | 48.7$\pm$3.0 |79.0 $\pm$ 1.9 |
> > |tGNN(ours)|84.4 $\pm$ 1.9|75.3 $\pm$ 2.0| 73.8 $\pm$ 0.9 | 49.1 $\pm$ 1.5| 82.1 $\pm$ 2.7|
> >
> > The table records graph classification results on test sets for 5 graph-level tasks under the setting of [3]. We can see that our model makes improvements on 4 out of 5 datasets (big improvements on MUTAG, PROTEINS, and COLLAB, and small improvements on IMDB-B).
> >
> > [1] Accurate Learning of Graph Representations with Graph Multiset Pooling. ICLR 2021.
> >
> > [2] Second-Order Pooling for Graph Neural Networks. IEEE Transactions on Pattern Analysis and Machine Intelligence, 2020.
> >
> > [3] A Fair Comparison of Graph Neural Networks for Graph Classification. ICLR 2020.

---

### Author Response · Authors · 2022-08-10
**A revision with more content**

Thanks for all your good suggestions which improve the paper quality and presentation. I have made a revision that includes our discussions. I manage to change some minor parts in the main paper (due to the page limitation) and write additional sections in the Appendix (D, E, F, G, H, I, J) including the fundamental questions and empirical results. I hope it doesn't bother you, but the revision does have a lot of good additional content. Thank you all.

---

### Meta-Review · Area_Chair_8XAT · 2022-08-26

**Recommendation:** Accept
**Confidence:** Certain

**Metareview:**

The reviewers initially disagree on whether this paper has sufficiently advanced the research topic of graph pooling and have concerns on missing the comparison with Wang and Ji (2020), which have also introduced higher-order pooling for graph neural networks (GNNs). After extensive discussions with the authors, the reviewers have reached a consensus towards acceptance: While the performance improvement is marginal, and in some cases, not statistically significant, the proposed tensor decomposition-based pooling algorithm is new and provides a theoretically-sound valuable addition to advanced neighborhood aggregation methods for GNNs.


**Award:**

No

---

### Decision · Program_Chairs · 2022-09-14

Accept